# Learning Binary Networks on Long-Tailed Distributions

## Abstract

In deploying deep models to real world scenarios, there are a number of issues including computational resource constraints and long-tailed data distributions. For the first time in the literature, we address the combined challenge of learning long-tailed distributions under the extreme resource constraints of using binary networks as backbones. Specifically, we propose a framework of calibrating off-the-shelf pretrained full precision weights that are learned on *non-long-tailed* distributions when training binary networks on long-tailed datasets. In the framework, we additionally propose a novel adversarial balancing and a multi-resolution learning method for better generalization to diverse semantic domains and input resolutions. We conduct extensive empirical evaluations on 15 datasets including newly derived long-tailed datasets from existing balanced datasets, which is the largest benchmark in the literature. Our empirical studies show that our proposed method outperforms prior arts by large margins, *e.g.*, at least $+14.33\%$ on average.

## 1 Introduction

In recent years, there grows emphasis on resource constraints, especially for edge devices, in learning deep models, resulting in breakthroughs such as MobileNet (Howard et al., 2017) and YOLO-V7 (Wang et al., 2022) that concern not only accuracy but also computational costs. This has attracted attention for deep learning both in the research and the industrial communities. Besides, long-tailed (LT) training data are frequently encountered in the wild (He & Garcia, 2009). Thus, many deep learning methods have been developed to combat it (Cui et al., 2021; He et al., 2021; Zhong et al., 2021). Yet, the devices of daily usages, of many which aim to utilize these deep models, suffer from lack of sufficient computing power. Thus, for real world deployment of such deep models, methodological advances in LT recognition should also consider the resource constraints.

Unfortunately, current LT recognition methods largely assume sufficient computing resources and are designed to work with a large number of full precision parameters, *i.e.*, floating point (FP) weights. While recognition on LT distributions by FP models may have significantly improved (Cui et al., 2021; He et al., 2021; Zhong et al., 2021), it is not clear whether these improvements would immediately translate to real world scenarios or not where resource constraints limit the model selection to those with lower capacity. To this end, we argue that it is necessary to benchmark and improve the performance of long-tailed recognition with capacity-limited models.

As binary networks are at the extreme end of capacity-limited models (Rastegari et al., 2016), the long-tailed recognition performance using the 1-bit networks would roughly correspond to the 'worst case scenario' for resource constrained LT. If we can show sufficient LT performance with binary networks, we can reasonably expect, at the very least, matching or better LT performance with $N$-bit models, where $N > 1$. Here, we take an initiative to benchmark and develop long-tailed recognition methods using binary networks as a challenging reference.

In the LT scenario, the data scarcity in the tail classes is one of the major issues that may cause problems such as the class weights having varying magnitudes for head to tail classes (Kang et al., 2019; Alshammari et al., 2022), leading to disappointing performance. Prior methods (Liu et al., 2019; Kozerawski et al., 2020; Park et al., 2022) use cosine classifiers which eliminate the effect of class weight norms as a discriminative statistic and improve accuracy. However, since binary networks lack learning capacity and exhibit worse generalization performance than FP networks (Rastegari et al., 2016; Courbariaux et al., 2016), we want to reduce the adverse effect of the uneven class

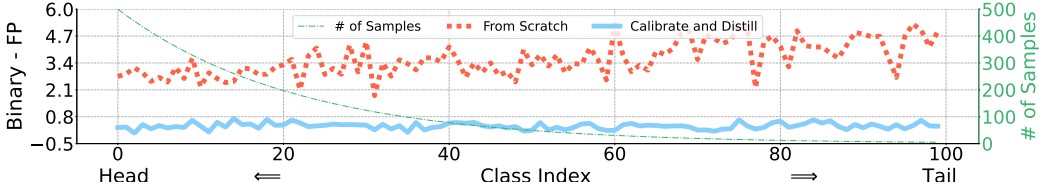

Figure 1: Gap between the classifier weight norms of binary and floating point (FP) network (*i.e.*, 'Binary-FP') trained using Adam on CIFAR-100 (imbalance ratio: 100) (Sec. 4), also used in Alshammari et al. (2022)). '# Samples' indicate the number of training samples for each class and the classes are sorted by the number of training samples (large to small), *e.g.*, head to tail. When trained from scratch on the long-tailed (LT) distribution, the gap widens towards the tail classes due to data scarcity. The widening gap implies that the binary network has higher variance (*i.e.*, larger weight magnitudes) compared to FP. Using our 'Calibrate and Distill' mitigates the gap substantially.

weight norms without sacrificing its usefulness as a discriminative statistic. Inspired by previous LT literature (Kang et al., 2019; Alshammari et al., 2022) that analyzed the classifier weight norms (of the last fully-connected layer) in FP networks with linear classifiers, we first investigate the high variance issue in binary networks (Zhu et al., 2019) by visualizing the Binary-FP gap in the magnitude of the weight norms of the classifier in Fig. 1. The widening gap towards the tail classes indicates that a binary network has higher variance at tail classes than FP models. We argue that we need to mitigate the above issue to improve LT recognition with binary networks.

One can reduce the high variance in binary networks by providing additional supervisory signals to the binary networks during training. The additional supervision would supplement the insufficient information due to the scarce data in the tail classes. In a different line of work, binary network learning methods (Martinez et al., 2020) employ additional supervision by a pretrained FP network used as a teacher in knowledge distillation when training binary networks. Furthermore, works in LT recognition with FP backbones also use distillation (He et al., 2021) to improve the accuracy.

However, these methods are not straightforwardly applicable to our scenario for the following reason. As most of pretrained FP networks are learned on non LT datasets, using them as teachers may not be helpful for training binary networks on LT datasets due to the domain gap. One can train the FP teacher on the target LT datasets from scratch (He et al., 2021) before distillation but that is not scalable. Recent works in various domains such as prompt engineering (Radford et al., 2021; Liu et al., 2021a) and self-supervised learning (Bachman et al., 2019; Chen et al., 2020) suggest utilizing pretrained weights is desirable for scalability (He et al., 2020). Here, we propose a way to utilize pretrained teacher networks on non LT data (*e.g.* from open source projects) for LT distillation *without* training the teacher from scratch. Specifically, we propose a 'Calibrate and Distill' framework where we calibrate the pretrained teacher on non LT data by attaching and training the classifier on the target LT dataset. We investigate how the proposed method affects the variance and visualize the results in Fig. 1. As shown in the figure, the calibrate and distill method significantly decreases the 'Binary-FP' gap, suggesting a reduction of the high variance in binary networks. Note that while the magnitudes are within similar magnitude ranges, they are not equal for all classes, which could be useful for discriminating different classes.

In addition, semantic domains and image resolutions vary across multiple real-world data. In addressing the variety, we found that balancing the amount of knowledge transferred from the distillation loss to the feature extractor and classifier part of the binary network largely affects the performance, depending on how varied the target data is to the pretraining data. However, manual tuning may result in good performance only for a limited number of benchmark datasets and may not generalize to various scenarios. For better generalization to different data distributions, we propose a novel adversarially learned balancing scheme for long-tailed recognition that learns the balance dependent on the input data without requiring finicky hyper-parameter tuning.

Furthermore, we incorporate various input resolutions for better generalization to different data distributions (Jacobs et al., 1995; Rosenfeld, 2013). This would ease the burden for the model to learn various optimal receptive fields that can help generalize to the differently sized data. But using multi-resolution inputs (Rosenfeld, 2013) may incur increasing computational costs depending on what and how many resolutions are used. For negligible training time increase in multi-scale learning, we propose to selectively use multi-resolution inputs to mitigate the extra computation costs

within our calibrate and distill framework. Combining all the proposals, we name our method as **Ca**librate a**n**d **D**istill: **L**ong-Tailed Recognition on the **E**dge or **CANDLE** for short.

We empirically evaluate on a wide variety of datasets totalling to 15, where some are derived by under-sampling existing non LT datasets and some are from existing LT benchmarks, that far outnumbers the previously used datasets in the LT literature. On our empirical validations in the various datasets, our method outperforms prior arts by large margin of at least $+14.33\%$ on average.

We summarize our contributions as follows:
- Addressing resource constrained long-tailed recognition with binary networks for the first time.
- Proposing a calibrate and distill framework for long-tailed recognition with binary networks utilizing the off-the-shelf pretrained teachers on non-LT distributions.
- Proposing a adversarially learned balancing scheme and efficient usage of multi-resolution inputs for the calibrate and distill framework.
- Evaluating the methods in a widely varied set of 15 long-tailed datasets for a comprehensive empirical study (the largest empirical study in the LT recognition literature).

## 2 RELATED WORK

**Long-tailed recognition.** As training deep learning models on LT distributions is challenging, numerous works have been proposed. We categorize them into class rebalancing (He & Garcia, 2009; Buda et al., 2018; Shen et al., 2016), logit adjustment (Cao et al., 2019; Cui et al., 2019; Tan et al., 2020), two stage methods – sequential learning of representation and classifier (Kang et al., 2019; Xiang et al., 2020; Ren et al., 2020; Tang et al., 2020; Li et al., 2020), and knowledge distillation (He et al., 2021; Wang et al., 2020a; Zhang et al., 2021a; Xiang et al., 2020).

Prior arts usually train the model from the scratch on the target LT dataset, which may be less efficient than using pretrained weights. For instance, two stage methods (Kang et al., 2019; Xiang et al., 2020; Ren et al., 2020; Tang et al., 2020; Li et al., 2020) perform representation learning without utilizing off-the-shelf pretrained weights. Granted, the pretrained weights are usually trained on a non-LT dataset with a considerable domain gap. However, we aim to utilize those pretrained weights on *non-LT* distributions for more efficient and effective long-tailed recognition.

Furthermore, some works tend to increase the model capacity for the long-tailed problem (Wang et al., 2020a; Zhou et al., 2020; Cui et al., 2021; 2022) which incurs additional costs. For the real-world scenario, one has to consider the resource constraints, thus resource efficient methods should be preferred. Additionally, most works are benchmarked only on a few datasets with little regards to actual data variety in the wild. To more accurately benchmark LT methods that consider the realistic variety, we use a total of 15 different LT datasets whereas previous works mostly showed results on 3-9 (Cao et al., 2019; Kang et al., 2019; Cui et al., 2021; 2022) datasets.

**Binary networks.** As one of the ways of bringing extreme efficiency to deep learning models, many prior arts have been proposed that either improves upon the underlying architecture of binary networks (Rastegari et al., 2016; Lin et al., 2017; Liu et al., 2018; 2020; Bulat et al., 2021), use neural architecture search to find better binary networks (Kim et al., 2020; Kim & Choi, 2021; Bulat et al., 2020), elaborate training schemes for effective optimization of binary networks (Martinez et al., 2020; Han et al., 2020; Meng et al., 2020; Liu et al., 2021b; Le et al., 2022), or explore using binary networks as backbones for previously unused scenarios such as unsupervised learning (Kim & Choi, 2022; Shen et al., 2021).

Recent successful attempts in training binary networks utilize a FP network to provide additional supervisory signals (Martinez et al., 2020; Liu et al., 2020; Bulat et al., 2021; 2020; Kim et al., 2020; Kim & Choi, 2021; 2022). However, these methods use teacher networks that are trained on data with similar distributions to that of the target data which amounts to little domain gap in the teacher network. As this process is not scalable, we propose to utilize teacher networks trained on *non-LT* data for many different target LT data distributions, which is scalable but challenging.

Additionally, most existing works focus on the supervised classification problem and methods that consider other scenarios such as unsupervised representation learning (Kim & Choi, 2022; Shen et al., 2021) and object detection (Wang et al., 2020b) are relatively scarce. As such, there is hardly any previous work regarding long-tailed recognition with binary networks which brings resource efficiency to learning on long-tailed distributions. Here, we aim to train resource efficient binary

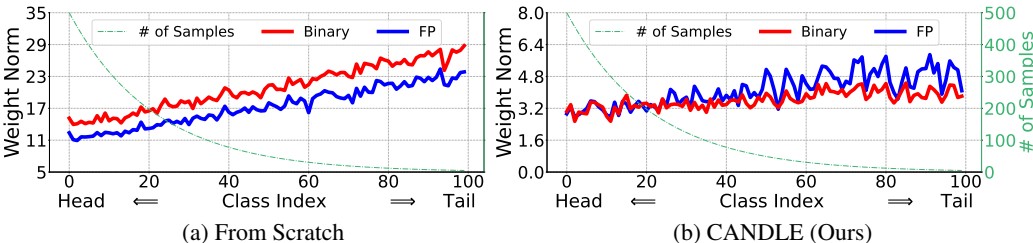

Figure 2: Classifier weight norms of binary and FP models trained with Adam on CIFAR-100 (imbalance ratio: 100). (a) The classifier weight norms increase at the tail classes, implying high variance. When trained from scratch, binary networks show larger weight norms than FP in tail classes, implying that the high variance issue is more severe in binary networks. (b) Using CANDLE, the weight norms of binary network become smaller than the FP model with little increase for tail classes. This implies that our method reduces the high variance in LT for binary networks.

networks that can also perform accurate long-tailed recognition to better prepare for long-tailed recognition in real-world deployment.

## 3 APPROACH

To develop long-tailed recognition methods for real world deployment that considers realistic resource constraints, we use binary networks as backbone networks, which are extremely efficient neural networks with single bit weights and activations (Rastegari et al., 2016) (see Sec. A.1 for a brief preliminary). We first explain how the class weight norms (Kang et al., 2019; Alshammari et al., 2022) can be used as an approximate to visualize the variance of a model and analyze its trends in LT (Sec. 3.1). With the analysis, we describe the proposed 'Calibrate and Distill' framework (Sec. 3.2) and plot the classifier weight norms to investigate how the proposed framework reduces the high variance in binary networks (Fig. 4). Additionally, to make the model more generalizable to the various semantic domains and image resolutions present in the wild that may differ from benchmark to benchmark, we further propose adversarially learned balancing (Sec. 3.3) and an efficient usage of multi-resolution inputs with negligible computational cost increase (Sec. 3.4).

### 3.1 HIGH VARIANCE FOR BINARY NETWORKS IN LONG-TAILED DISTRIBUTION

The scarcity of data for tail classes impacts the bias and variance of a model when it is trained on LT distributions (Wang et al., 2020a). Specifically, we believe the model with linear classifiers is likely to exhibit high variance as it fits to the few training samples in the tail classes. We hypothesize that the above will be more severe for binary networks as they have limited capacity, making the models more susceptible to overfitting to the scarce training data in the tail. Previous works (Kang et al., 2019; Alshammari et al., 2022) analyzed the imbalance in the classifier weight norms for classes ranging from head to tail. We carefully re-visit how the classifier weight norm analysis could be used in our scenario as there are a number of differences that might affect the analysis as follows.

First, we found that previous works (Kang et al., 2019; Zhou et al., 2020; Alshammari et al., 2022) used the SGD optimizer with weight-decay when analyzing the classifier weight norm. However, using the Adam optimizer significantly outperforms using the SGD optimizer when training binary networks (Han et al., 2020). Thus, using SGD will lead to inaccurate observations as the analysis will be done for models that have not sufficiently fit to the training data yet.

Second, weight decay is a regularization technique that pushes the weights to be closer to zero, which is not desirable for binary networks (Han et al., 2020) which quantize the weights to $+1/-1$. Furthermore, as weight decay regularizes the training, it may result in the binary network underfitting to the training data, similar to using the SGD optimizer (Han et al., 2020; Kim & Choi, 2021). Perhaps even more problematic, weight decay may artificially reduce the weight magnitudes, making accurate analysis of the variance challenging. Thus, we use the Adam optimizer with zero weight decay in observing the classifier weight norms for our experiments Please see Sec. A.10 for more discussion on the difference in weight norm trend with Adam and SGD.

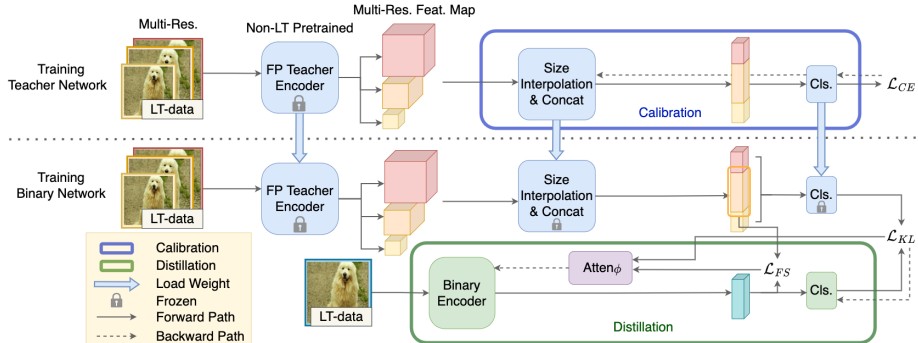

Figure 3: Overview of the proposed CANDLE. The FP classifier is calibrated with multi-resolution inputs. In distillation, only the teacher network receives multi-resolution inputs which saves computational costs significantly (see Fig. 5). Adversarially learned balancing via 'Atten$_\phi$' is used in training the binary encoder. During calibration, the FP teacher encoder is frozen. During distillation, the FP teacher encoder, size interpolation and concat module, and the FP classifier are frozen.

We visualize the classifier weight norms in Fig. 2-(a). When the model is trained from scratch, we observe that the classifier weight norms get bigger as the classes become more tail. A similar phenomenon can be observed in machine learning literature where the model is trained to have high variance (*e.g.*, high weight magnitudes) when it fits excessively to the training data (Kohavi et al., 1996; Von Luxburg & Schölkopf, 2011). Thus, one can interpret the above observation as a signal of overfitting to the scarce training samples in the tail, leading to the weights for the tail classes having high magnitudes and large variance (see Sec. A.2 for more details). In line with our hypothesis, this trend is intensified for binary networks. As shown in Fig. 2-(a), the binary networks exhibit consistently larger weights with widening gap between the binary and floating point models (see Fig. 1) towards the tail classes.

In contrast, we apply the proposed CANDLE as depicted in Fig. 3 and conduct the same analysis using classifier weight norms in Fig. 2-(b). The magnitudes of the binary weights are largely reduced with the gap between the binary and FP being reduced significantly (as also shown in Fig 1). We attribute the reduction in variance to the 'Calibrate and Distill' component of CANDLE that supplies additional supervisory signals in the tail classes from the FP teacher.

## 3.2 Calibrate and Distill

To overcome the high variance issue, we propose a 'Calibrate and Distill' scheme that provides additional information to the binary network during training over the given scarce training data in the tail classes. Inspired by recent methods in training binary networks that use a FP teacher for distillation (Martinez et al., 2020), we use pretrained FP networks as distillation teacher networks in training the binary network on LT data. Note that most existing pretrained weights are not necessarily trained on LT distributions and it is also not scalable to manually acquire pretrained weights on each target LT datasets. Here, we propose the 'Calibrate and Distill' scheme to utilize non-LT pretrained weights for use on LT datasets in an efficient manner.

Specifically, we propose to first calibrate the teacher network with non-LT pretrained weights and then transfer the knowledge from the calibrated teacher. In particular, we attach a randomly initialized classifier to the non-LT pretrained FP network. During calibration, the classifier is trained using the cross entropy loss $\mathcal{L}_{CE}$ on the target LT datasets using Zhong et al. (2021), which is more efficient than training the entire teacher. We use the KL divergence and feature similarity losses between the calibrated teacher and the binary networks for distillation following (Romero et al., 2014; Heo et al., 2019). The optimization objective is

$$\min_{\theta} \mathbb{E}_{x \sim \mathcal{D}_{LT}}[(1-\lambda)\mathcal{L}_{KL}(f^T(x), (f_\theta^B(x)) + \lambda\mathcal{L}_{FS}(e^T(x), e_\theta^B(x))], \tag{1}$$

where $\mathcal{D}_{LT}$ is the target LT dataset and $f^T(\cdot), f_\theta^B(\cdot)$ are the calibrated teacher and binary networks, respectively. $e^T(\cdot), e_\theta^B(\cdot)$ are the teacher and binary encoders, *i.e.*, feature extractors, and $\mathcal{L}_{KL}(\cdot, \cdot), \mathcal{L}_{FS}(\cdot, \cdot)$ are the KL divergence and cosine distance, inspired by Chen et al. (2020); He et al. (2020), respectively.

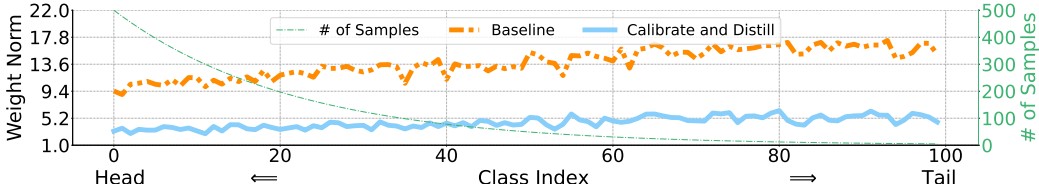

Figure 4: 'Calibrate and Distill' *vs.* the baseline of using the pretrained weights without calibration in terms of classifier weight norms on CIFAR-100 (imbalance ratio: 100). 'Calibrate and Distill' maintains the magnitudes relatively small even for tail classes whereas the baseline does not, showing the effectiveness of 'Calibrate and Distill' in reducing the high variance compared to the baseline.

When we empirically compare our 'Calibrate and Distill' to the baseline, *i.e.*, using the pretrained weights without calibrating with feature distillation (Romero et al., 2014; Heo et al., 2019), we see an accuracy gain of $+8.75\%$ on average (see Tab. 3). More interestingly, observing the classifier weight norms we found that the proposed 'Calibrate and Distill' method is highly effective in mitigating the high variance compared to the baseline as shown in Fig. 4-(a). Note that while the high variance in binary networks is reduced through 'Calibrate and Distill', varying semantic domains with diverse image resolutions in the real world data are not sufficiently addressed yet.

## 3.3 ADVERSARIALLY LEARNED BALANCING

The encoder of our teacher network is pretrained on non-LT data but the classifier is trained on the target LT datasets. Optimizing the binary encoder and the classifier altogether with Eq. 1 will disregard this difference and may disrupt the training of the binary encoder by using the gradients from $\mathcal{L}_{KL}$ in the same way that we use it to train the binary classifier. Thus, depending on how much the LT target data differs from the non-LT pretraining data, we may employ different strategies for balancing $\mathcal{L}_{FS}$ and $\mathcal{L}_{KL}$ via $\lambda$ to train the binary encoder and the classifier. However, manually tuning $\lambda$ for $\mathcal{L}_{KL}$ and $\mathcal{L}_{FS}$ for each different target data is not desirable for generalization.

To address the semantic diversity without manual tuning, we propose to parameterize and learn $\lambda$ during training as well. To that end, we first split the binary network's parameters $\theta$ into the binary encoder parameters ($\theta_e$) and classifier parameters ($\theta_c$), and optimize them separately as

$$
\min_{\theta_c} \mathbb{E}_{x \sim \mathcal{D}_{LT}}[\mathcal{L}_{KL}(f^T(x), f^B_{\theta_c}(x))],
$$
$$
\min_{\theta_e} \mathbb{E}_{x \sim \mathcal{D}_{LT}}[(1 - \lambda)\mathcal{L}_{KL}(f^T(x), f^B_{\theta_e}(x)) + \lambda \mathcal{L}_{FS}(e^T(x), e^B_{\theta_e}(x))].
$$
(2)

To learn $\lambda$, we use $\mathcal{L}_{KL}$ and $\mathcal{L}_{FS}$ as inputs and use an attention module with learnable parameters $\phi$ to output a single scalar value $\lambda_\phi$ such that $\lambda_\phi$ is dependent on the two loss functions as

$$
\lambda_\phi = \text{Atten}_\phi(\mathcal{L}_{KL}(x), \mathcal{L}_{FS}(x)),
$$
(3)

where $\text{Atten}_\phi(\cdot)$ is a MLP that takes in $\mathcal{L}_{KL}(x)$ and $\mathcal{L}_{FS}(x)$, and outputs $\lambda_\phi$. Note that minimizing with respect to $\phi$ results in the learned $\lambda_\phi$ outputting zero as the balancing coefficient for whichever loss function is larger *i.e.*, 0 for $\mathcal{L}_{FS}$ in Eq. 2 if $\mathcal{L}_{FS} > \mathcal{L}_{KL}$. While this does *minimize* the loss, it also prevents the gradient flow from $\mathcal{L}_{FS}$ when optimizing with respect to $\theta$. Thus, we employ adversarial learning where $\lambda_\phi$ is learned to maximize the loss which is then minimized by optimizing $\theta$. We can rewrite our objective as

$$
\min_{\theta_c} \mathbb{E}_{x \sim \mathcal{D}_{LT}}[\mathcal{L}_{KL}(f^T(x), f^B_{\theta_c}(x))],
$$
$$
\min_{\theta_e} \max_{\phi} \mathbb{E}_{x \sim \mathcal{D}_{LT}}[(1 - \lambda_\phi)\mathcal{L}_{KL}(f^T(x), f^B_{\theta_e}(x)) + \lambda_\phi \mathcal{L}_{FS}(e^T(x), e^B_{\theta_e}(x))].
$$
(4)

## 3.4 LEARNING WITH MULTI-RESOLUTION INPUTS

To make our model more robust to various image resolutions frequent in the wild, we argue that using multi-resolution inputs during training would be a plausible solution (Rosenfeld, 2013). Specifically, we propose to use $128 \times 128$, $224 \times 224$ and $480 \times 480$ resolution images as multi-resolution inputs, following Tan & Le (2019). However, directly using multi-resolution inputs dramatically increases

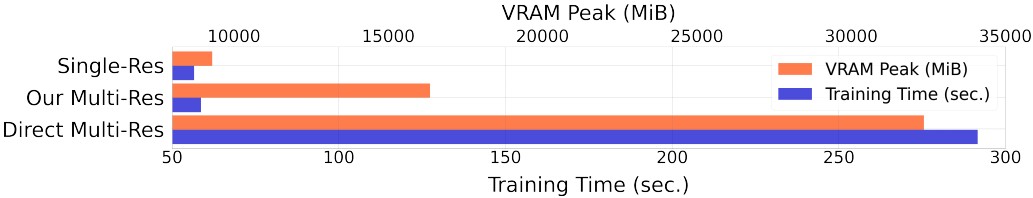

Figure 5: Computational (blue bars) and memory (orange bars) costs comparison between directly using multi-resolution inputs to our method of using them only for the teacher ('Our Multi-Res') in terms of VRAM peak (MiB) and 1 epoch training time (sec.). 'Our Multi-Res' shows negligible increase in training time compared to using single resolution. While the VRAM peak does increase with 'Our Multi-Res', it is still far more efficient than directly using multi-resolution inputs.

the training time as shown in Fig. 5 (Single-Res *vs.* Direct Multi-Res). Thus, we propose a novel way to efficiently use multi-resolution inputs for our 'Calibrate and Distill' framework such that there is negligible increase in training time (Single-Res *vs.* Our Multi-Res in Fig. 5).

As we harness supervisory signals from both the input data and the teacher network, we can select which supervisory signal source the multi-resolution inputs are used. Instead of using multi-resolution inputs in both sources, we only use them when obtaining supervisory signals from the teacher. To do so, we first calibrate the teacher using multi-resolution inputs as shown in Fig. 3. As the calibration only trains the classifier (see Sec. 3.2), the usage of multi-resolution inputs is relatively inexpensive. During distillation, we feed the multi-resolution inputs to the teacher only and use single-resolution inputs for the binary network. Because the teacher is not part of the backward computational graph, the increase in training time is negligible (see Fig. 5). The 'VRAM peak' indicates the peak consumption of GPU memory during training. Our scheme does incur some VRAM peak increase but is much more efficient than training schemes that directly use the multi-resolution inputs to both teacher and binary networks. Note that with the multi-resolution inputs, the feature similarity loss is calculated using only the channels that correspond to the input with the same resolution as that of the input given to the binary network as depicted in Fig. 3.

We use calibrate and distill with the adversarially learned balancing and an efficient way of using multi-resolution inputs as our full method named as **CANDLE**.

## 4 EXPERIMENTS

**Setup.** For empirical evaluations, we use 12 small-scale and 3 large-scale LT datasets as detailed in Sec. A.4. We use pretrained FP networks from public open source projects that are either trained on ImageNet-1K for the small-scale and on ImageNet-21K for the large-scale LT datasets. Following (Liu et al., 2021b), we use the Adam optimizer with zero weight decay for training binary networks. Further implementation details are in Sec. A.5. Note that we use the authors' implementation of prior arts when available or re-implement by reproducing the reported results. More details on the compared prior arts are in Sec. A.3. We will publicly release the code for CANDLE along with the newly derived LT datasets.

**Extended benchmarks for long-tail recognition.** In addition to developing a method for LT recognition using binary networks, we also want to improve how the performance of LT methods are measured to better reflect the real-world scenario (Yang et al., 2022). To that end, we compile a large number of derived LT datasets from commonly used computer vision datasets along with the datasets often used in the LT literature to gather 12 small-scale and 3 large-scale datasets for evaluation. We denote the imbalance ratio in a parenthesis following the dataset names for the small-scale datasets and add a postfix of '−LT' to each of the large-scale dataset names. More details regarding on the newly added datasets along with dataset statistics for all 15 datasets are given in Sec. A.4.

### 4.1 LONG-TAILED RECOGNITION ACCURACY IN VARIOUS DATASETS

We present comparative results of our method with various state of the arts LT methods tuned for binary networks. We show the average accuracy over multiple datasets. Please see Sec. A.11 for head, medium, and tail class accuracy and per-class accuracy plots for CANDLE. All results of CANDLE are with linear classifiers (see Sec. A.12 for results with cosine classifiers on small-scale

Table 1: Average accuracy (%) of the proposed CANDLE and other methods on small-scale datasets. We compare with $\tau$-norm (Kang et al., 2019), BALMS (Ren et al., 2020), PaCo (Cui et al., 2021), MiSLAS (Zhong et al., 2021), and DiVE (He et al., 2021). DiVE* is trained using the same teacher as ours. Our method shows superior performance by large margins across many different datasets with a margin of at least $+14.90\%$ in mean accuracy.

| Dataset (Imb. Ratio) | $\tau$-norm | BALMS | PaCo | MiSLAS | DiVE | DiVE* | **CANDLE** |
|---|---|---|---|---|---|---|---|
| Caltech101 (10) | 20.68 | 17.00 | 45.85 | 36.38 | 20.91 | 39.66 | **55.31** |
| CIFAR-10 (10) | 24.28 | 17.40 | 58.59 | 82.25 | 13.73 | 79.46 | **91.76** |
| CIFAR-10 (100) | 21.72 | 16.50 | 61.57 | 69.50 | 16.26 | 58.41 | **85.01** |
| CIFAR-100 (10) | 8.18 | 2.70 | 49.63 | 47.03 | 2.26 | 49.93 | **66.09** |
| CIFAR-100 (100) | 5.56 | 3.40 | 39.66 | 32.31 | 1.94 | 32.95 | **50.88** |
| CUB-200-2011 (10) | 8.52 | 1.30 | 15.41 | 14.74 | 4.09 | 12.82 | **42.94** |
| Stanford Dogs (10) | 13.93 | 9.80 | 34.04 | 29.73 | 6.72 | 29.21 | **58.79** |
| Stanford Cars (10) | 7.73 | 2.90 | 23.74 | 24.82 | 4.49 | 19.31 | **51.06** |
| DTD (10) | 16.86 | 14.10 | 36.65 | 36.54 | 12.71 | 30.37 | **38.56** |
| FGVC-Aircraft (10) | 7.71 | 4.30 | 15.12 | 22.08 | 1.95 | 14.19 | **39.78** |
| Flowers-102 (10) | 45.29 | 43.60 | 52.16 | 60.20 | 23.82 | 58.53 | **64.61** |
| Fruits-360 (100) | 85.82 | 97.30 | 99.23 | 99.49 | 50.87 | 99.71 | **100.00** |
| Mean Acc. | 22.23 | 19.19 | 44.30 | 46.14 | 13.31 | 41.80 | **62.07** |

datasets). The best accuracy is in **bold** and the second best is underline. Surprisingly, DiVE (He et al., 2021) does not perform well on the small-scale datasets as the teacher network used in DiVE's distillation process, *e.g.*, BALMS (Ren et al., 2020) exhibits disappointing accuracy ($\sim 26.65\%$ mean acc.) in the first place. Hence, we also present results with DiVE* which uses the *same* teacher network as our method. More discussion regarding some of the other low performing methods on the small-scale datasets such as $\tau$-norm (Kang et al., 2019) and BALMS is in Sec. A.6.

**In small-scale datasets.** We summarize the comparative results on the 12 small-scale datasets in Tab. 1 along with the mean accuracy over all 12 datasets. The mean accuracy over the 12 datasets suggests that CANDLE performs better than the prior arts by noticeable margins of at least $+15.93\%$. More precisely, the proposed method outperforms all other existing works by large margins for the popular CIFAR-10 (10,100) and CIFAR-100 (10,100) datasets. In addition, our method exhibits superior performance in some of the newly added LT datasets with limited data such as CUB-200-2011 (10). Interestingly, comparing DiVE and DiVE*, we can see that using our calibrated teacher boosts the performance significantly. However, the performance difference between DiVE* and CANDLE ($+20.27\%$) also indicates that our adversarially learned balancing (Sec. 3.3) and multi-resolution teacher is effective (Sec. 3.4). Additional results on changing the teacher network to the larger one used for large-scale experiments (Sec. 4) is in Sec. A.7, which show that the mean accuracy improves when using the larger teacher. Further comparison against the ensemble-based method, TADE (Zhang et al., 2021b), is also in Sec. A.8.

**In large-scale datasets.** Similar to Tab. 1, we compare our method to prior arts on the large-scale datasets in Tab. 2. While our multi-resolution teacher is far more efficient than directly using multi-resolution (Fig. 5), the VRAM peak still increases. Thus, we could not run the multi-resolution experiments with the large-scale datasets with our resources and only use single-resolution, indicated using † in Tab. 2. Still, even without using multi-resolution, the proposed method outperforms prior arts with binary networks on all three large-scale datasets. Looking at the mean accuracy, our method shows a margin of at least $+7.93\%$ when compared to the previous LT works.

## 4.2 LEARNED $\lambda_\phi$ IN DIFFERENT DATASETS

The learned values of $\lambda_\phi$ (Eq. 4) for each dataset *vs.* training iteration is visualized in Fig. 6. Note that we do not schedule $\lambda_\phi$ in any specific way. Interestingly, we observe smooth transitions from the initial values to 1 for all the shown datasets. The initial values are different as they depend on the different initial loss values for each dataset. In addition, while the learned $\lambda_\phi$ converges to 1 (using only the $\mathcal{L}_{FS}$) in Eq. 4, the way in which the $\lambda_\phi$ changes is different depending on the dataset.

Combined with the empirical gain ($+5.40\%$ mean average accuracy in Tab. 3) of using adversarially learned balancing, the various different ways in which $\lambda_\phi$ varies in Fig. 6 imply that adversarially learned balancing (Eq. 4) is effective in balancing $\mathcal{L}_{KL}$ and $\mathcal{L}_{FS}$ on a wide variety of LT datasets.

Table 2: Average accuracy (%) of CANDLE and other methods on large-scale datasets. $^\dagger$ indicates that we only use single-resolution due to VRAM limits. DiVE$^*$ is trained using the same teacher network as ours. The proposed method outperforms other existing works in all three large-scale datasets with at least $+7.93\%$ margin in terms of mean accuracy.

| Dataset | $\tau$-norm | BALMS | PaCo | MiSLAS | DiVE | DiVE$^*$ | **CANDLE**$^\dagger$ |
|---|---|---|---|---|---|---|---|
| Places-LT | 25.99 | 23.60 | 23.30 | 28.49 | 20.96 | 22.93 | **34.11** |
| ImageNet-LT | 30.59 | 33.00 | 34.58 | 34.71 | 31.21 | 30.86 | **49.10** |
| iNat-LT | 37.36 | 44.30 | 45.73 | 43.59 | 41.32 | 41.30 | **47.38** |
| Mean Acc. | 31.31 | 33.63 | 34.54 | 35.60 | 31.16 | 31.70 | **43.53** |

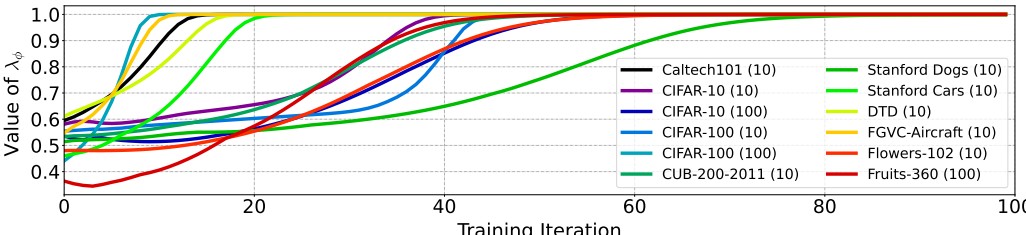

Figure 6: Learned value of $\lambda_\phi$ on different datasets during training by Eq. 4. Note that the convergence behavior of $\lambda_\phi$ to $1.0$ differ for each dataset.

Table 3: Ablation studies of our method, CANDLE. We present the mean accuracy over the 12 small-scale datasets (Tab. 4). Using our 'Calibration & Distill' framework drastically improves the performance from the baseline of using the pretrained weights without calibration. Adversarially learned balancing and multi-resolution inputs also improve the mean accuracy substantially.

| Method | ① Calibrate & Distill | ② Adv. Learned Bal. | ③ Multi-Res. | Accuracy (%) | | | |
|---|---|---|---|---|---|---|---|
| | | | | Avg. | Head | Med. | Tail |
| Baseline | ✗ | ✗ | ✗ | 45.27 | 53.55 | 44.40 | 32.47 |
| ① | ✓ | ✗ | ✗ | 55.59 | 62.99 | 55.90 | 42.57 |
| ①+② | ✓ | ✓ | ✗ | 60.99 | 66.64 | 61.72 | 49.61 |
| ①+②+③ (=CANDLE) | ✓ | ✓ | ✓ | **62.07** | **71.22** | **63.33** | **50.09** |

## 4.3 ABLATION STUDIES

We finally conduct ablation studies of our ① 'Calibrate & Distill', ② 'Adv. Learned Bal.', and ③ 'Multi-Res.' on the 12 small-scale datasets and report the mean average, head, medium, and tail class accuracy in Tab. 3. The baseline is set to using the uncalibrated teacher with feature distillation (Romero et al., 2014; Heo et al., 2019) (refer to Sec. 3.2 for detail). Note that the baseline achieves $45.27\%$ mean average accuracy, only behind MiSLAS. All proposed components of CANDLE improve the mean average accuracy, e.g., $+10.32\%$ for 'Calibrate & Distill', $+5.40\%$ for 'Adv. Learned Bal.', and $+1.08\%$ for 'Multi-Res.'. Furthermore, the mean tail class accuracy also improves consistently with each component, indicating the effectiveness of the respective components for LT. The average accuracy for each dataset for the ablated models can be found in Sec. A.9.

## 5 CONCLUSION

To develop efficient deep learning models for LT, we use binary networks as backbones for learning long-tailed recognition. In doing so, we analyze a high variance issue and propose to mitigate it using our 'Calibrate and Distill' framework where pretrained weights on non-LT data are utilized as distillation teachers for learning binary networks on LT target data distributions. We further propose adversarially learned balancing and efficient usage of multi-resolution inputs to better prepare the model for the wide variety of semantic domains and input resolutions present in the wild. We empirically evaluate the proposed CANDLE and other existing works on a total of 15 datasets, which is the largest benchmark in the literature. The proposed method achieves superior performance in all empirical validations with ablations showing clear benefits of each component of CANDLE.

ETHICS STATEMENT

This work aims to equip edge-compatible binary networks with long-tailed recognition capabilities. Thus, though there is no intent on the authors, application of deep models to real-world scenarios with long-tail data distributions might become prevalent. This may expose the public to discrimination by the deployed deep models due to unsolved issues in deep learning such as model bias. We will take all available measures to prevent such outcomes as that is *not* the intention of this work.

REPRODUCIBILITY STATEMENT

We take the reproducibility in deep learning very seriously and highlight some of the contents in the manuscript that might help in reproducing our work. First, we will release the code and newly derived datasets used in our experiments as mentioned in Sec. 4. Second, we include additional information regarding the extended long-tailed benchmark in Sec. A.4 that may help with future work looking to use the datasets that were used in this work. Third, we include relevant implementation details in both Sec. 4 and Sec. A.5. Last, we present our final optimization objective (Eq. 4) and the overall view of our method (Fig. 3) with necessary details to reproduce the methodology.

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

## A    APPENDIX

### A.1    PRELIMINARY ON LEARNING BINARY NETWORKS (SEC. 3)

Let $\mathbf{W_F} \in \mathbb{R}^{C_i \times C_o \times k \times k}$ be the 32-bit floating point weights of a convolution layer where $C_i, C_o$ and $k$ represent the number of input channels, the number of output channels and the kernel size, respectively. The corresponding binarized weights are given by $\mathbf{W_B} = \mathrm{sign}(\mathbf{W_F})$, where the $\mathrm{sign}(\cdot)$ outputs $+1/-1$ depending on the sign of the input. The floating point input activation $\mathbf{A_F} \in \mathbb{R}^{C_i \times w \times h}$, where $w, h$ are the width and height of the input activation, can be similarly binarized to the binary input activation by $\mathbf{A_B} = \mathrm{sign}(\mathbf{A_F})$. A floating point scaling factor is also used, which can either be the L1-norm of $\mathbf{W_F}$ (Rastegari et al., 2016) or learned by back-propagation (Bulat & Tzimiropoulos, 2019).

The forward pass using convolutions is approximated as following

$$\mathbf{W_F} * \mathbf{A_F} \approx \alpha \odot (\mathbf{W_B} \oplus \mathbf{A_B}), \tag{5}$$

where $*, \odot, \oplus$ denote the floating point convolution operation, Hadamard product, and the binary XNOR convolution with popcount, respectively. STE (Courbariaux et al., 2016) is used for the weight update in the backward pass.

### A.2    DETAILED ANALYSIS ON THE HIGH VARIANCE (SEC. 3.1)

We give a more detailed explanation on the high variance analysis in Sec. 3.1. As we are more interested in the high weight magnitudes which signals high variance, we focused on showing the classifier weight norms in the main part of the paper. Here, we also show the bias portion of the classifier and see how it is affected for binary networks in LT conditions.

As seen in Fig. 7-(a), we see the classifier weight norm (*e.g.* variance) become larger and larger for the tail classes when the binary network is trained from scratch. Looking at the bias in Fig. 7-(b), we see the classifier bias getting smaller and smaller for the tail classes accordingly when the binary network is trained from scratch. In contrast, when CANDLE is applied instead, the large increase in the classifier weight norm for the tail classes is substantially mitigated, resulting in the weight norm barely changing from head to tail classes. Similarly, the bias is also relatively unchanging compared to training from scratch and shows that CANDLE stabilizes the binary network in terms of both variance and bias.

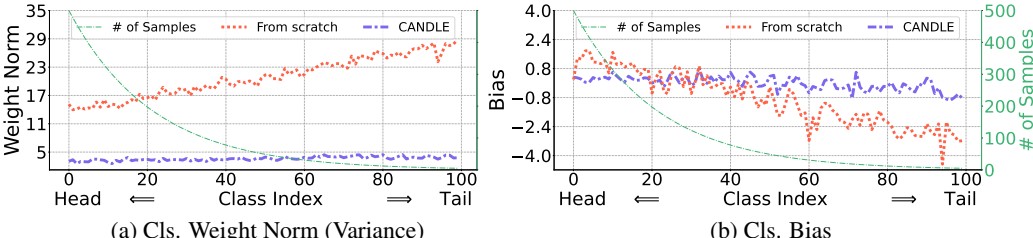

(a) Cls. Weight Norm (Variance)          (b) Cls. Bias

Figure 7: Classifier weight norm (variance) and bias for binary networks trained from scratch or using CANDLE on CIFAR-100 (100). We can see that training from scratch results in binary networks with large variance and small bias whereas using CANDLE, binary networks have relatively stable variance and bias.

### A.3    DETAILS ON COMPARED METHODS (SEC. 4)

We give brief summary of the compared methods. $\tau$-norm (Kang et al., 2019) is a two-stage de-coupling method, which consists of classifier re-training(cRT) and learnable weight scaling(LWS). BALMS(Ren et al., 2020) is also a two-stage method, where the balanced softmax loss and meta sampler is used in classifier training stage. PaCo(Cui et al., 2021) uses supervised contrastive learning to learn the class centers in long-tailed data distributions. MiSLAS(Zhong et al., 2021) focuses on calibrating the model properly on LT data with a two-stage method that uses CutMix(Yun et al., 2019) to enhance the representation learning stage. The classifier learning stage consists of label-aware smoothing and shift learning on batch normalization layers. DiVE(He et al., 2021) is a distil-

lation method where the teacher network is trained by existing well perfroming LT-methods such as BALMS and then trains student model with KL divergence loss and balanced softmax loss.

## A.4 Details on the Extended LT Benchmark (Sec. 4)

Table 4: Dataset statistics of all the LT datasets used in our experiments. The dataset source denotes the original non-LT dataset from which the LT datasets were derived. The imbalance ratio for the large scale datasets are not explicitly calculated.

| Dataset | Source | Imbalance Ratio | # Train Samples | # Test Samples |
|---|---|---|---|---|
| | Caltech101 | 10 | 1151 | 6084 |
| | CIFAR-10 | 10 | 20431 | 10000 |
| | CIFAR-10 | 100 | 12406 | 10000 |
| | CIFAR-100 | 10 | 19573 | 10000 |
| | CIFAR-100 | 100 | 10847 | 10000 |
| | CUB-200-2011 | 10 | 2253 | 5794 |
| Small Scale | Stanford Dogs | 10 | 4646 | 8580 |
| | Stanford Cars | 10 | 3666 | 8041 |
| | DTD | 10 | 1457 | 1880 |
| | FGVC-Aircraft | 10 | 2564 | 3333 |
| | Flowers-102 | 10 | 753 | 1020 |
| | Fruits-360 | 100 | 1770 | 3110 |
| | Places-LT | N/A | 62500 | 7300 |
| Large Scale | ImageNet-LT | N/A | 115846 | 50000 |
| | iNat-LT | N/A | 437513 | 24426 |

the newly added LT datasets from existing computer vision datasets *e.g.*, Caltech101 (Li et al., 2022), CUB-200-2011 (Wah et al., 2011), Stanford Dogs (Khosla et al., 2011), Stanford Cars (Krause et al., 2013), DTD (Cimpoi et al., 2014), FGVC-Aircraft (Maji et al., 2013), Flowers-102 (Nilsback & Zisserman, 2008), and Fruits-360 (Mureşan & Oltean, 2017). We first sort the class indices with the number of samples per class to create a class order. We then set the imbalance ratio and sub-sample the data in each of the classes. In sub-sampling the data, we use an exponential decaying curve to determine the class sample frequency from head to tail classes. As the number of data in each of the classes is limited, we often can only use a certain imbalance ratio such as 10 or 100.

We then detail the dataset statistics for all the LT datasets used in our experiments in Tab. 4, including the newly added ones and the previously used datasets in the LT literature. The added datasets have a wide variety of semantic domains which may have not been covered by the existing LT datasets in the literature.

## A.5 Further Implementation Details (Sec. 4)

In the calibration stage, we used the class-balanced sampling and label-aware smoothing following MiSLAS (Zhong et al., 2021). We use a pretrained ResNet-152 (He et al., 2015) backbone for the ImageNet1K teacher and a pretrained EfficientNet (Tan & Le, 2019) backbone for the ImageNet21K teacher. In distillation stage, we used ReActNet (Liu et al., 2020) as our binary backbone network. $\text{Atten}_\phi$ is a MLP with 3 hidden layers with the number of channels expanding by 16 per layer starting at 2 channels in the first layer. The MLP takes two scalar values $\mathcal{L}_{KL}(x)$ and $\mathcal{L}_{FS}(x)$ turned into a 2-dimensional vector and outputs a single scalar value which is passed through a sigmoid function to be used as $\lambda_\phi$. To process the multi resolution inputs to the teacher network, we use the size interpolation & concat module. The size interpolation & concat module is comprised of 1x1 convolutions that match the number of channels to that of the binary network and a nearest interpolation step where the different sized feature maps from the multi-resolution inputs are spatialy resized. Once the feature maps from the different resolution inputs have the same spatial size and number of channels, they are channel-wise concatenated. To optimize the min-max objective of Eq. 4, we perform 1 maximization step per every 1 minimization step. The binary networks are trained for 400 epochs with a batch size of 256. We use the cosine annealing scheduling for our

Table 5: Accuracy (%) of the proposed CANDLE with different teacher networks. CANDLE uses the ImageNet1k teacher as detailed in Sec. 4 and CANDLE[‡] uses the ImageNet21K teacher which was originally used for large-scale experiments in Tab. 2.

| Datasets (Imb. Ratio) | CANDLE (ImageNet1K Teacher) | CANDLE[‡] (ImageNet21K Teacher) |
|---|---|---|
| Caltech101 (10) | 55.31 | 52.27 |
| CIFAR-10 (10) | 91.76 | 91.27 |
| CIFAR-10 (100) | 85.01 | 85.59 |
| CIFAR-100 (10) | 66.09 | 64.93 |
| CIFAR-100 (100) | 50.88 | 53.15 |
| CUB-200-2011 (10) | 42.94 | 49.45 |
| Stanford Dogs (10) | 58.79 | 63.51 |
| Stanford Cars (10) | 51.06 | 49.73 |
| DTD (10) | 38.56 | 40.64 |
| FGVC-Aircraft (10) | 39.78 | 41.64 |
| Flowers-102 (10) | 64.61 | 66.27 |
| Fruits-360 (100) | 100.00 | 100.00 |
| Mean Acc. | 62.07 | 63.20 |

Table 6: Accuracy (%) of the proposed CANDLE with TADE (Zhang et al., 2021b) which is an ensemble-based method. Our method shows superior performance against even ensemble-based TADE by large margins across many different datasets with a margin of $+24.94\%$ in mean accuracy.

| Dataset (Imb. Ratio) | TADE | CANDLE |
|---|---|---|
| Caltech101 (10) | 43.86 | **55.31** |
| CIFAR-10 (10) | 53.54 | **91.76** |
| CIFAR-10 (100) | 47.63 | **85.01** |
| CIFAR-100 (10) | 29.75 | **66.09** |
| CIFAR-100 (100) | 28.68 | **50.88** |
| CUB-200-2011 (10) | 13.00 | **42.94** |
| Stanford Dogs (10) | 17.20 | **58.79** |
| Stanford Cars (10) | 8.73 | **51.06** |
| DTD (10) | 27.98 | **38.56** |
| FGVC-Aircraft (10) | 12.69 | **39.78** |
| Flowers-102 (10) | 62.46 | **64.61** |
| Fruits-360 (100) | 100.00 | **100.00** |
| Mean Acc. | 37.13 | **62.07** |

learning rate scheduler, where the initial learning rate is set as 0.01 for most of the datasets. Although we did try different learning rates, we found the difference to be negligible. We used one GPU for training on the 12 small-scale datasets and four GPUs for training on the 3 large-scale datasets.

## A.6 MORE DISCUSSION REGARDING $\tau$-NORM AND BALMS (SEC. 4.1)

On comparison with existing LT methods on the small-scale datasets in Tab. 1, $\tau$-norm and BALMS seems to show relatively low accuracy. We hypothesize the following reasons for the low accuracy. For $\tau$-norm, it is based on how the weight norms distributed when the model is trained with the SGD optimizer. Since we use the Adam optimizer to train the models due to having binary backbones (Sec. 3.1), the core assumption for $\tau$-norm may not hold and hence the resulting low accuracy. For BALMS, it may have lacked the sufficient number of epochs to properly converge as we use binary backbone networks which usually take longer to train than floating point models (Courbariaux et al., 2015).

## A.7 EFFECT OF USING DIFFERENT TEACHER NETWORKS FOR CANDLE (SEC. 4.1)

We used the ImageNet1k pretrained teacher network for small-scale experiments in Tab. 1 as we felt that the teacher network could still provide sufficient supervisory signals for the small-scale datasets.

Nonetheless, we also conduct experiments on the small-scale datasets with the larger teacher network pretrained on ImageNet21k which was originally used only for the large-scale datasets.

As shown in Tab. 5, not all datasets benefit from the larger teacher network such as Caltech101 (10), CIFAR-10 (10), CIFAR-100 (10), and Stanford Cars (10). However, there are noticeable gains of $+6.51\%$ on CUB-200-2011-10 and $+4.72\%$ on Stanford Dogs-10 and the resulting mean accuracy is improved by $+1.13\%$. For the large-scale datasets in Tab. 2, we could not use the ImageNet1K teacher as that is a direct superset of ImageNet-LT. The above reason is why we used the ImageNet21K teacher for the large-scale experiments. However, the gains in Tab. 5 suggest that one could use the ImageNet21K teacher for the small-scale datasets as well.

## A.8   ADDITIONAL COMPARISON TO TADE (ZHANG ET AL., 2021B) (SEC. 4.1)

For a more comprehensive comparison, we also present results for TADE (Zhang et al., 2021b), which is an ensemble-based method. As shown in Tab. 6, we show results for TADE on the 12 small-scale datasets along with CANDLE. Our method still outperforms TADE by large margins of $+24.94\%$ in terms of mean accuracy. Not only that, the proposed method seems to show higher accuracy both on the frequently used CIFAR-10 (10, 100) and CIFAR-100 (10, 100) datasets but also on the newly added datasets such as CUB-200-2011 (10), Stanford Dogs (10), and Stanford Cars (10).

## A.9   MORE DETAILED ABLATION STUDY RESULTS (SEC. 4.3)

We presented the mean accuracy over the 12 small-scale datasets for our ablation studies in Tab. 3 to more clearly present the performance of the ablated models. We also report the accuracy on each of the 12 datasets in Tab. 7 to provide more details regarding our ablation studies. However, we also emphasize that looking at the accuracy of a particular dataset only may not give the most accurate description.

Going from ① to ① + ② + ③ (=CANDLE), 9 out of 12 datasets show an accuracy increase. On the 3 datasets where an accuracy decrease is observed, the accuracy drops by $-5.17\%$ at most, whereas on 9 datasets where the accuracy increases, it increases by up to $+41.16\%$. Ultimately, the mean accuracy consistently goes up when add more and more parts of CANDLE, as also shown in Tab. 3.

Table 7: A more detailed ablation studies of our method CANDLE. The mean accuracy over 12 small-scale datasets as well as the accuracy for each dataset are shown. The circled numbers denote the following ①: 'Calibrate & Distill', ②: Data-Driven Bal., ③: Multi-Res., respectively. Using our 'Calibration & Distill' framework drastically improves the performance across all the tested datasets when compared to the baseline of using the pretrained weights without calibration. 'Data-Driven Bal.' and 'Multi-Res.' also improve the accuracy substantially over all the tested datasets.

| Datasets (Imb. Ratio) | Baseline | Ablated Models | | |
|---|---|---|---|---|
| | | ① | ① + ② | ① + ② + ③ (=CANDLE) |
| Caltech101 (10) | 36.75 | 53.44 | 53.37 | 55.31 |
| CIFAR-10 (10) | 88.90 | 82.08 | 92.34 | 91.76 |
| CIFAR-10 (100) | 70.00 | 43.85 | 87.24 | 85.01 |
| CIFAR-100 (10) | 48.96 | 69.07 | 68.58 | 66.09 |
| CIFAR-100 (100) | 33.62 | 54.92 | 53.70 | 50.88 |
| CUB-200-2011 (10) | 12.69 | 37.66 | 37.61 | 42.94 |
| Stanford Dogs (10) | 25.66 | 63.96 | 64.42 | 58.79 |
| Stanford Cars (10) | 25.87 | 44.00 | 45.19 | 51.06 |
| DTD (10) | 21.33 | 40.21 | 36.38 | 38.56 |
| FGVC-Aircraft (10) | 28.02 | 29.91 | 29.22 | 39.78 |
| Flowers-102 (10) | 51.86 | 62.16 | 63.82 | 64.61 |
| Fruits-360 (100) | 99.61 | 85.85 | 100.00 | 100.00 |
| Mean Acc. | 45.27 | 55.59 | 60.99 | 62.07 |

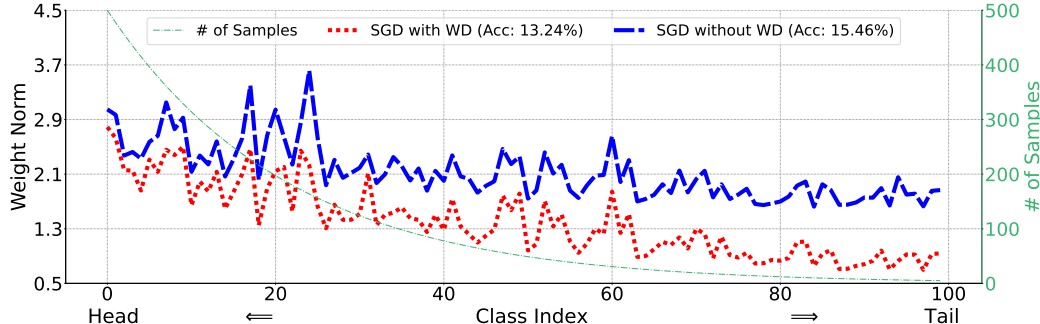

Figure 8: Classifier weight norm of binary networks trained from scratch on CIFAR-100 (100) using SGD with or without weight decay. Using the SGD, we get similar trends for the class weight norms as in Kang et al. (2019); Alshammari et al. (2022) where the class weight norms get smaller for the tail classes. However, as mentioned in Sec. 4.2, using the SGD when training binary networks results in very low accuracy of $13.24\%$ or $15.46\%$ depending on whether weight decay is used or not. Thus, we use the Adam optimizer, in which the classifier weight trends are different (see Sec. 3.1).

## A.10  MORE DISCUSSION ON DIFFERENCE IN WEIGHT NORM TREND WITH ADAM OR SGD

As discussed in Sec. 3.1, we use the Adam optimizer for training the binary networks. This results in a different trend in the classifier weight norms than some prior works (Kang et al., 2019; Alshammari et al., 2022). To give a more information regarding this matter, we also plot the classifier weight norm trend of training binary networks from scratch on CIFAR100 (100) using the SGD optimizer in Fig. 8. We see that using the SGD shows similar trends in terms of the classifier weight norms as the prior arts (Kang et al., 2019; Alshammari et al., 2022). However, the accuracy of the learned binary networks with SGD is very low for both with and without weight decay. We believe that further developing our analysis on such an under-fitted model will not be meaningful and use Adam optimizer instead.

Note that combined with results in Fig. 2, we see opposite trends in weight norms depending on whether we use Adam or SGD for both binary and FP networks. With Adam, we have weight norms increasing as the classes become tail, and with SGD, we have weight norms decreasing as the classes become tail instead. We conjecture that Adam adjusts the learning rates such that parameters corresponding to frequently occurring head class data use relatively small learning rates and parameters corresponding to infrequent tail class data use relatively large learning rates. As a result, the optimization may result in the tail class weights having large weight norms due to the relatively larger learning rate used in the update. Note that more thorough studies on the above phenomenon is warranted to gain a deeper understanding and we leave that as an interesting future direction, respecting the original scope of this work.

## A.11  ADDITIONAL INFORMATION REGARDING HEAD, MEDIAN, AND TAIL CLASS ACCURACY

To provide more information regarding the experimental results of CANDLE, we provide average, head, medium, and tail class accuracy of CANDLE for the 15 datasets in Tab. 8. For most of the datasets, CANDLE shows high accuracy at the tail classes compared to the average, head, or the medium classes. This is especially true for large-scale datasets such as Places-LT, ImageNet-LT, and iNat-LT where the tail class accuracy is either on par or higher than the average accuracy.

Additionally, in Fig. 9, 10, 11, we plot a per-class accuracy improvement of CANDLE over the baseline used in Tab. 3 following Kozerawski et al. (2020) on Places-LT, ImageNet-LT, and iNat-LT, respectively. The figures show that CANDLE improves the accuracy at the tail and medium classes by large margins for all three datasets. Additionally, in the case of ImagetNet-LT, even the head accuracy shows noticeable improvements. The figures empirically show that CANDLE shows good

Table 8: Average, head, medium, and tail class accuracy of CANDLE. On most datasets, CANDLE shows high tail class accuracy compared to the average, head, and medium class accuracy. Interestingly, this trend is stronger for large scale datasets where the tail class accuracy is higher or on par with the average accuracy.

|  | Dataset | Average | Head | Medium | Tail |
|---|---|---|---|---|---|
| Small Scale | Caltech101 (10) | 55.31 | 49.25 | 46.74 | 35.54 |
|  | CIFAR-10 (10) | 91.76 | 95.50 | 90.50 | 91.03 |
|  | CIFAR-10 (100) | 85.01 | 93.90 | 81.90 | 82.67 |
|  | CIFAR-100 (10) | 66.09 | 75.64 | 67.20 | 54.00 |
|  | CIFAR-100 (100) | 50.88 | 73.94 | 54.11 | 19.07 |
|  | CUB-200-2011 (10) | 42.96 | 50.94 | 40.13 | 26.30 |
|  | Stanford Dogs (10) | 58.79 | 67.06 | 57.09 | 45.35 |
|  | Stanford Cars (10) | 51.06 | 67.65 | 48.84 | 37.09 |
|  | DTD (10) | 38.56 | 59.82 | 41.32 | 26.61 |
|  | FGVC-Aircraft (10) | 39.78 | 29.55 | 51.05 | 42.04 |
|  | Flowers-102 (10) | 64.61 | 91.39 | 81.11 | 41.33 |
|  | Fruits-360 (100) | 100.00 | 100.00 | 100.00 | 100.00 |
| Large Scale | Places-LT | 34.11 | 34.62 | 33.83 | 37.41 |
|  | ImageNet-LT | 49.10 | 53.89 | 42.21 | 45.66 |
|  | iNat-LT | 43.53 | 48.94 | 41.96 | 46.35 |

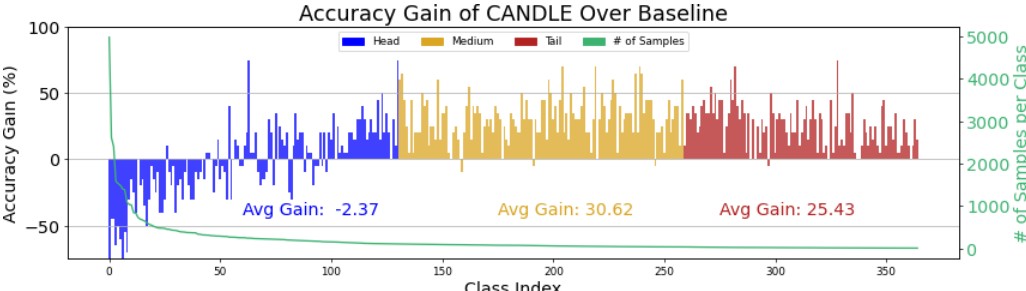

Figure 9: Per-class accuracy gain of CANDLE over the baseline on Places-LT. Our method improves the accuracy at the tail classes by $+25.43\%$, showing its effectiveness for LT. Additionally, the proposed method also improves the accuracy at the medium classes by $+30.62\%$ and only drops by $-2.37\%$ for the head classes.

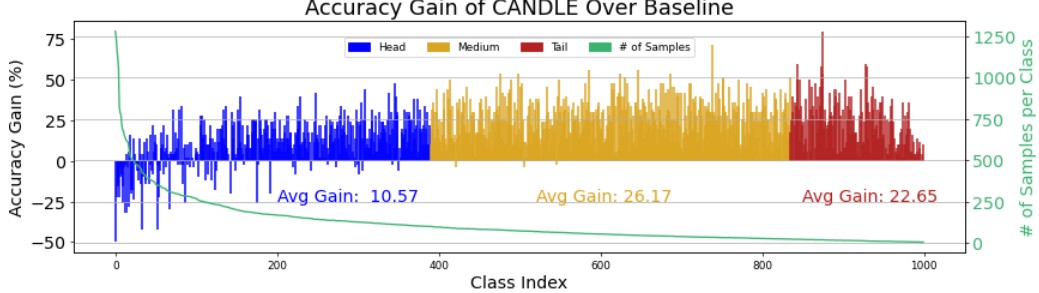

Figure 10: Per-class accuracy gain of CANDLE over the baseline on ImageNet-LT. Out method improves the accuracy at the tail classes by $+22.65\%$, showing its effectiveness for LT. Furthermore, the proposed method also shows gains in the accuracy at the head and medium classes by $+10.57\%$ and $+26.17\%$ respectively.

average accuracy not because it is only good at head classes but mostly because it improves the accuracy of tail and medium classes by large margins.

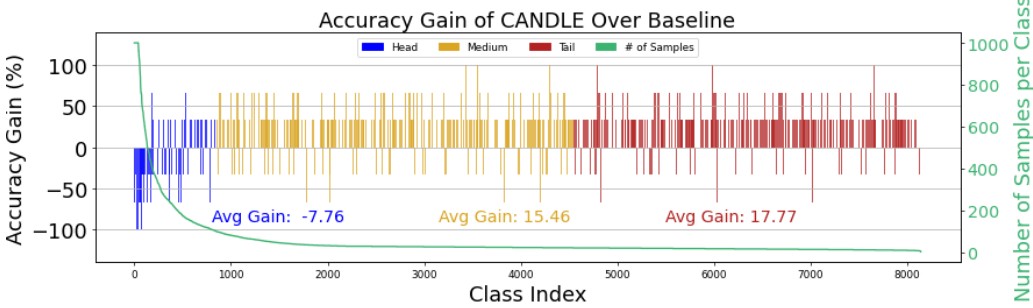

Figure 11: Per-class accuracy gain of CANDLE over the baseline on iNat-LT. Our method improves the accuracy at the tail classes by $+17.77\%$, showing its effectiveness for LT. The medium class accuracy is also improved by $+15.46\%$. There is a slight reduction in head class accuracy of $-7.76\%$.

| Dataset | CANDLE w/ Cosine Cls. | | | | CANDLE w/ Linear Cls. | | | |
|---|---|---|---|---|---|---|---|---|
| | Average | Head | Medium | Tail | Average | Head | Medium | Tail |
| Caltech101 (10) | 55.52 | 50.99 | 46.28 | 22.92 | 55.31 | 49.25 | 46.74 | 35.54 |
| CIFAR-10 (10) | 91.27 | 94.97 | 89.00 | 90.60 | 91.76 | 95.50 | 90.50 | 91.03 |
| CIFAR-10 (100) | 81.57 | 95.17 | 80.32 | 69.63 | 85.01 | 93.90 | 81.90 | 82.67 |
| CIFAR-100 (10) | 53.92 | 71.28 | 55.20 | 30.83 | 66.09 | 75.64 | 67.20 | 54.00 |
| CIFAR-100 (100) | 40.11 | 71.56 | 36.34 | 5.59 | 50.88 | 73.94 | 54.11 | 19.07 |
| CUB-200-2011 (10) | 39.13 | 51.54 | 39.49 | 24.76 | 42.96 | 50.94 | 40.13 | 26.30 |
| Stanford Dogs (10) | 56.74 | 66.53 | 55.82 | 40.03 | 58.79 | 67.06 | 57.09 | 45.35 |
| Stanford Cars (10) | 38.88 | 55.84 | 32.41 | 21.11 | 51.06 | 67.65 | 48.84 | 37.09 |
| DTD (10) | 40.85 | 61.25 | 38.82 | 23.21 | 38.56 | 59.82 | 41.32 | 26.61 |
| FGVC-Aircraft (10) | 30.03 | 25.03 | 41.36 | 22.58 | 39.78 | 29.55 | 51.05 | 42.04 |
| Flowers-102 (10) | 65.98 | 90.83 | 73.33 | 27.33 | 64.61 | 91.39 | 81.11 | 41.33 |
| Fruits-360 (100) | 99.87 | 100.00 | 99.63 | 100.00 | 100.00 | 100.00 | 100.00 | 100.00 |
| Mean Acc. (%) | 57.82 | 69.58 | 57.33 | 39.88 | 62.07 | 71.22 | 63.33 | 50.09 |

Table 9: Average, head, medium, and tail class accuracy of CANDLE with either cosine classifiers or linear classifiers on 12 small-scale datasets. Using linear classifiers almost always outperforms using cosine classifiers in all average, head, medium, and tail class accuracy except for results on Caltech101 (10).

## A.12   RESULTS WITH COSINE CLASSIFIERS FOR CANDLE ON SMALL-SCALE DATASETS

We present results of the proposed method with the cosine classifiers instead of the linear classifier in Tab. 9. As shown in the table, using linear classifiers results in better average accuracy than cosine classifiers for all datasets except Caltech101 (10). Not only is the average accuracy higher for linear classifiers, but all head, medium, and tail class accuracy show clear advantages of linear classifiers over cosine classifiers on all 12 datasets. Taking the mean over the 12 datasets shows a $+4.25\%$ gain in average accuracy as well as a $+10.21\%$ gain in tail class accuracy. This is in-line with our hypothesis in Sec. 1 that cosine classifiers may not work as well for binary networks due to the reduced usefulness of weight norms magnitudes in supplementing the limited capacity of binary networks.

## A.13   LIMITATIONS.

We mainly focus on the algorithmic aspects of learning binary networks with accurate LT recognition capabilities. Actual deployment of LT capable binary networks to the real-world requires more application-oriented development on top of the proposed algorithmic improvements and is out of scope of this work. Additionally, while our usage of multi-resolution inputs incurs little increase

in training time, an increase in the VRAM peak is observed. Reducing the VRAM peak increase would be an interesting future direction for making the algorithm more efficient.

