# OpenReview forum: "Learning Binary Networks on Long-Tailed Distributions"
_ICLR.cc/2023/Conference — Submitted to ICLR 2023_

### Official Review · Reviewer_2EyL · 2022-10-18

**Confidence:** 5
**Correctness:** 3
**Technical Novelty And Significance:** 4
**Empirical Novelty And Significance:** 4
**Recommendation:** 8

**Clarity, Quality, Novelty And Reproducibility:**

Clarity: this paper is easy to understand.
Quality: the quality is high but there are several questions that can be resolved.
Novelty: the novelty is high.
Reproducibility: authors are expected to release the source code to the public.


**Strength And Weaknesses:**

Strengths:
1. Learning binary networks on long-tailed data is a novel and practical task. Overall, I like this idea.
2. The analysis is interesting, which shows the differences between learning binary networks and learning float-point networks on long-tailed data.
3. Empirical results are encouraging.

Weaknesses:
1. Figure 2 finds that the classifier weight norms increase at the tail classes on both float-point networks and binary networks. However, this is contrary to several classic methods, like decoupling (Kang, ICLR 2020). They find classifier weight norms of float-point networks are positively related to long-tailed class distribution with standard training.  The authors explain that conventional FP methods use SGD and weight decay but binary networks use ADAM with zero weight decay. It makes sense, but it will be better to provide a baseline that binary networks use SGD and weight decay to demonstrate this.
2. Moreover, it is interesting to know the size of feature space per class. In conventional FP networks, head classes usually have large feature space than tail classes. How about binary networks?
3. Furthermore, many long-tailed approaches use the cosine classifier since its classifier weights are normalized. It is good to show how the proposed method performs with the cosine classifier.
4. Why do you use MiSLAS to calibrate the classifier? Efficiency seems not a superiority of MiSLAS.



**Summary Of The Paper:**

This paper studies long-tailed learning for binary networks. This is a novel setting. To address this, this paper proposes a new CANDLE method, which is based on three main strategies: (1) Calibrate and Distill, (2) adversarial learned balancing, (3) multi-resolution training. Promising empirical results demonstrate its effectiveness.

**Summary Of The Review:**

Despite a few questions that can be resolved, I like this paper as it explores a new and practical task, i.e., learning binary networks on long-tailed datasets, and provides a few insights. I tend to accept this paper.

---

> ### Author Response · Authors · 2022-11-09
> **Answers to the questions of Reviewer 2EyL**
>
> > Finding in Fig. 2 is contradictory to the previous ones by Kang et al., ICLR 2020 (using SGD). The provided explanation makes sense but it would be better to provide results for binary networks with SGD and weight decay for clear comparison.
>
> $\to$ We first thank the reviewer for appreciating our explanation for the difference in the weight-norm trend from previous work. We agree that showing the results using SGD with weight decay for binary networks would help the explanation. So, we present a plot of the weight norms for binary networks trained from scratch on CIFAR-100 (100) using SGD with and without weight decay in A.10 of the revision. Notice that the trend now matches that of the previous works with the usage of weight decay making the weight norm decrease towards the tail classes even more. But SGD does not perform well in optimizing binary networks (see Sec. 3.1); we obtain accuracy of only $13.24\%$ with weight decay and $15.46\%$ without weight decay compared  to $50.88\%$ using Adam without weight decay. Thus, while using SGD does result in a similar trend to previous works, the low accuracy of the trained model suggests that further analysis with such an under-fitted model might not be very fruitful.
>
>
> > In conventional FP networks, head classes usually have larger feature space than tail classes. How about binary networks?
>
> $\to$ Great point to investigate! We are currently working on calculating the feature space size and will add it to the following revision. We would greatly appreciate it if the reviewer could also point towards specific references for how to compute the feature space size in LT as the definition of the feature space size might vary for each researcher.
>
>
> > Good to show the results for the cosine classifier since its weights are normalized.
>
> $\to$ Great suggestion! We agree that results with cosine classifiers would be interesting and will add the results in the following revisions. Meanwhile, we want to cordially point out that binary networks have limited capacity and worse generalization than FP networks, which motivates us to not use cosine classifiers as they normalize the weights which may reduce the usefulness of the weight norms (the magnitude) as a discriminative statistic. We have added a discussion on the cosine classifier in the introduction of the current revision.
>
>
> > Why MiSLAS to calibrate the teacher? Efficiency is not a strength of MiSLAS.
>
> $\to$ Yes, the efficiency is not the superiority of MiSLAS. The main reason we use MiSLAS to calibrate the teacher is its superior empirical performance, as can be observed in Tab.1 and Tab.2. Note that we only use MiSLAS to retrain the classifier part of the teacher, not the entire network. Thus, the computational cost of using MiSLAS is much less than that of its normal usage of training the entire network. Additionally, as CANDLE can be used with other methods in the teacher calibration stage, a more efficient and effective LT method can definitely be used instead of MiSLAS as well.

---

> > ### Comment · Reviewer_2EyL · 2022-11-18
> > **Further discussion**
> >
> > Thanks for the response! The revision has been improved a lot. I have one more question. When you use SGD to train binary networks, the weight norm trend becomes consistent with the FP networks. That is, ADAM plays a key role in the weight norm trend. Can you also train a FP network with ADAM on CIFAR100-LT, and see whether it becomes the same trend with binary networks with ADAM regarding weight norms?

---

> > > ### Author Response · Authors · 2022-11-18
> > > **Further discussion with reviewer 2EyL**
> > >
> > > >  Can you also train a FP network with ADAM on CIFAR100-LT, and see whether it becomes the same trend with binary networks with ADAM regarding weight norms?
> > >
> > > $\to$ Thanks for the interesting suggestion. The blue line in Fig. 2 - (a) shows the weight norms of a FP network trained with Adam on CIFAR100-LT (which we call CIFAR-100 (imbalance ratio: 100)) that show a similar trend with binary networks, *i.e.*, becoming larger for the tail classes. Note that the trend is also observed with binary networks with an amplified effect; for instance, Fig. 1 shows the ‘Binary - FP’ gap of the weight norms when trained with Adam on CIFAR-100 (100). There, the *gap* itself also becomes larger when it approaches the tail classes, implying that binary networks have bigger weight norms for tail classes than FP networks. We have revised the captions in Figures 1 and 2 to clearly indicate that the networks are trained with Adam.
> > >
> > > > That is, ADAM plays a key role in the weight norm trend.
> > >
> > > $\to$ We thank the reviewer for bringing up this interesting observation. We agree with the reviewer that the results may imply that Adam plays an important role in the weight norm trend for LT. We speculate that the adaptive nature of Adam may result in relatively small learning rates for head class data as they are frequently observed in training and large learning rates for tail class data as they appear less frequently in training than the head class samples. This may result in large tail class weight norms due to the large learning rates used in the update. We have added this brief discussion on the matter in A.10 of the third revision but leave more thorough investigations on this subject as future work in respect to the scope of the original manuscript.

---

> > > > ### Comment · Reviewer_2EyL · 2022-11-19
> > > > **Response to Authors**
> > > >
> > > > Thanks a lot for your response, which clearly addresses my concerns. The quality of this paper is further improved. I keep my rating to 8 (acceptance) and improve my confidence to 5. Good luck.

---

> ### Author Response · Authors · 2022-11-15
> **Follow-up answers to the questions of Reviewer 2EyL**
>
> **Update on our answer to your question of**
> > In conventional FP networks, head classes usually have larger feature space than tail classes. How about binary networks?
>
> $\to$ We calculated the maximum distance of the feature vectors to their respective class centers for the head and tail classes. We observe that the trend regarding the size of the feature space for head classes (19.27) is bigger than that of the tail classes (18.93) for binary networks as well, albeit the difference being relatively small. Please let us know if the reviewer would like more details on this matter.
>
> \
> **Update on our answer to your question of**
> > Good to show the results for the cosine classifier since its weights are normalized.
>
> $\to$ As suggested, we add the results of using cosine classifiers instead of linear classifiers for CANDLE in A.12 of the second revision. The results show that the linear classifiers are advantageous. Please see the general response and the second revision for more details.

---

### Official Review · Reviewer_ah6w · 2022-10-23

**Confidence:** 2
**Correctness:** 3
**Technical Novelty And Significance:** 3
**Empirical Novelty And Significance:** 3
**Recommendation:** 3

**Clarity, Quality, Novelty And Reproducibility:**


I think the clarity can be improved, and consequently this can improve the reproducibility of the paper. However, I think the idea of using an adversarial learning framework is interesting.

**Strength And Weaknesses:**

Strengths:
1. The motivation of the paper has a practical impact. Tackling long-tail scenarios is an important problem in real applications, and especially, considering the computational efficiency aspect is an important problem.
2. The idea of introducing a learned weight (\lambda) that controls the “focus” between the two loss-terms in order to guarantee that the maximum loss at every iteration is minimized is interesting. I think the learning formulation in the teacher-student framework is a new idea.
3. Overall, the clarity of the intuition and high-level ideas are explained well. However, there are some concerns I have with details and motivation; see the section below.

Weaknesses:
1. Assumptions are not well justified, and the paper is not self-contained:
- The main assumption from the paper is that a neural-network-based image classifier suffers from performance issues due to the weight-norm variances. This already assumes that the networks use a linear-based classifier. However, previous work has shown that cosine-based classifiers also improve the performance of image classifiers in the long-tail scenario; see references A and B. In theory, these cosine-based classifiers should be more robust to the weight-norm variance because only angles are the main discriminative statistic.
- The paper is not self-contained because it lacks a full discussion about the bad impact the weight-norm variance has when learning from long-tailed datasets. The reader has to dig in deeper and read extra to understand this claim.
- Moreover, the narrative in the introduction projects to the reader that the main cause of the bad performance when learning from long-tailed datasets is the weight-norm variance. However, as shown in previous work (see references A and B for example), the lack of data from tail classes is the main cause of the problem. Thus, over sampling methods also alleviate the bad performance effect when learning from a long-tailed dataset. I think the introduction needs a revision in which it acknowledges previous work in a more structured and inclusive manner of previous work.

2. The paper lacks clarity:
- It is unclear from the introduction and Section 3.2 how the teacher network is pre-trained on non-LT data. What does this mean? Does it mean the teacher network used a balanced dataset (e.g., CIFAR 100), and then the teacher transfers its knowledge to the student network? I could not find a clear explanation about this.
- In section 3.1, the paper states that it is believed that the “model is likely to exhibit high [weight-norm] variance as it fits to the few training samples in the tail classes”. I think this statement needs to be backed up with an experiment.

3. Insufficient experiments: In the long-tail context, the mean average accuracy is not very informative. This is because it is unclear if the head classes are improving and thus moving the average accuracy metric up. I think a plot showing accuracy performance per class can be more informative (see Fig. 4 in A). It is unclear if the method favors more head or medium-shot classes. Note that this also applies to the ablation study.

4. While CANDLE can improve efficiency during the inference stage due to the use of binary-based networks, the training of these networks is quite elaborate. In my (humble) opinion, this decreases the practical benefit as it requires training a teacher network first, then training a student network using CANDLE again.

References:

A. Kozerawski, et al.. BLT: Balancing Long-Tailed Datasets with Adversarially-Perturbed Images. ACCV 2020.

B. Park, et al.. The Majority Can Help the Minority: Context-Rich Minority Oversampling for Long-Tailed Classification. CVPR 2022



================= Post-Discussion =================

After engaging in a discussion with the authors, I will downgrade my recommendation to 3 - Reject, not good enough. The reason for this is the following: the proposed approach is not practical. To me, the proposed approach has a fundamental problem. The proposed approach assumes that a teacher model (i.e., a non-LT model) exists and can be used to train a student model that will effectively deal w/ the long-tail dataset. In a realistic scenario and as an example, if I want to train a classifier using CANDLE and the Google Open Images dataset, which is a long-tailed one, it would be very hard to get a non-LT teacher model. Therefore, I cannot use CANDLE in large scale and real long-tailed dataset. Therefore, I question the approach as a viable solution to the long-tailed problem. Unfortunately, I could not get a satisfactory answer to this concern.

**Summary Of The Paper:**

The paper presents CANDLE, a binary-based network trained using a teacher-student framework in which a learned weight selects the largest loss between two loss terms in the optimization problem. In this way, CANDLE ensures the minimization of the maximum loss at every iteration. The motivation of using a binary network is to ensure a simple but efficient network that can be used in long-tail scenarios with efficiency in mind. Also, the paper adds a multi-resolution training schema in order to boost performance. The paper presents experiments on synthetically long-tailed and natural long-tail dataset demonstrating that the average accuracy improves considerably.

**Summary Of The Review:**

Overall, I think the paper lacks clarity, better justifications of their algorithmic design choices, and more informative experiments showing the performance on head, medium-shot, and tail classes. However, I am intrigued about the adversarial learning framework applied to the teacher and student paradigm.

---

> ### Author Response · Authors · 2022-11-09
> **Answers to the questions of Reviewer ah6w**
>
> > Assumption of using linear-based classifiers with no discussion on cosine-based classifiers.
>
> $\to$ We agree that using cosine classifiers is another way to handle the large class weight norms as it can largely eliminate the impact of the class weight norms. But it is not clear if this is as beneficial for binary networks as well. The reason is that since binary networks have limited capacity and worse generalization than FP networks (Rastegari et al., 2016; Courbariaux et al., 2016), using cosine classifiers may not utilize the differences in class weights norms to be used as a discriminative statistic, which may hurt performance. Thus, we want to address the large class weight norms with binary networks in a way that does not reduce its potential usefulness. We have clarified this in the introduction of the revised manuscript by including a discussion on the cosine classifiers and why we are inclined to not use them.
>
> Nonetheless, we are also curious about the performance of CANDLE if we replace the linear classifier used in CANDLE with a cosine based one and will add the results in the following revisions.
>
>
> > More discussion on some of the other problems in LT such as data scarcity as well as more details on why the weight-norm variance is problematic for long-tailed datasets, e.g., for self-containedness.
>
> $\to$ Thank you for a good suggestion! We completely agree that the data scarcity in the tail classes is a main problem in LT. In fact, we argue that because of such data scarcity, the problem of fitting to a small number of training data is worsened in the LT scenario, which may result in large class weight norms, *i.e.*, the weight-norm variance problem. We have revised the introduction and  Sec. 3.1 to more clearly address this.
>
>
> > How is the teacher trained on non-LT data?
>
> $\to$ We use publicly available pretrained weights from open source projects such as PyTorch or PyTorch Image Models which are pretrained on non-LT datasets such as ImageNet1K and ImageNet21K. We have revised the introduction and Sec. 4 for better clarity.
>
>
> > In Sec 3.1, the claim “model is likely to exhibit high weight-norm variance as it fits to the few training samples in the tail classes” needs experimental results.
>
> $\to$ Thank you for the suggestion! We cordially refer the reviewer to Fig 2-(a) which shows the weight-norm (variance) for binary and floating point networks when trained from scratch on long-tailed datasets. The class weight norms have increasing magnitudes towards the tail classes, which empirically supports the questioned claim.
>
>
> > Results lack per class accuracy plot and detailed accuracy for head, medium, and tail classes.
>
> $\to$ Great suggestions! We have added head, medium, and tail class accuracy to the ablation studies in Tab. 3 of the revision. Furthermore, the head, medium, and tail classes for the proposed CANDLE on all 15 datasets have been added to Tab. 8 in A.11 of the revision. Finally, we also add the per-class accuracy improvement of CANDLE over the baseline for the three large-scale datasets, (Places-LT, ImageNet-LT, iNat-LT) in Fig. 9, 10, 11 in A.11 of the revision.
>
> As shown in the updated ablation studies, the proposed components of CANDLE help improve the tail class accuracy, indicating their effectiveness for LT. The added results in Tab. 8 show that CANDLE shows high tail class accuracy, which is sometimes *on par or higher* than the average accuracy for certain datasets such as Places-LT, ImageNet-LT, and iNat-LT. The per-class accuracy improvements in Fig. 9, 10, 11 show that CANDLE improves largely in the tail and medium class accuracy over the baseline, with the exception of ImageNet-LT where CANDLE improves upon the head class accuracy as well. The added results collectively show that CANDLE shows not only high average accuracy but also high tail class accuracy which is important for LT scenarios.
>
>
> > While CANDLE can improve efficiency during the inference stage, training seems quite expensive as it needs to train the teacher network first then train a student using CANDLE. Thus, it may decrease the practical benefits of CANDLE.
>
> $\to$ We appreciate that the reviewer acknowledges that efficiency during inference of CANDLE is improved due to the use of binary networks. Furthermore, we cordially argue that the practical benefit of CANDLE is more on the inference cost and not the training cost. In addition, our method only incurs little cost for using a teacher network as we do **not** train the teacher network ourselves but utilize pretrained off-the-shelf weights from various open source projects. We only need to calibrate the teacher (retrain just the classifier which is inexpensive) on the target LT dataset before training the binary network.

---

> > ### Comment · Reviewer_ah6w · 2022-11-13
> > **RE: Answers to the questions of reviewer ah6w**
> >
> > Thanks for the clarifications, I am satisfied with most of the points except with one concern. The fact that the proposed method uses a teacher pre-trained model on a non-LT dataset is very concerning. This is because I find all the comparisons with the baselines presented in the experiment sections *unfair*. This is because clearly the unbiased teacher will provide useful prior information to the student especially on tail classes. This extra prior of information is not present in the baselines which makes the comparisons unfair. I think this may explain why a simpler and faster network (i.e., the binary networks) are getting good numbers presented in the experiments. Another concern is that in practice (or in the real world) having a non-LT dataset to train a teacher and then use the proposed method is very challenging. Thus I think the impact of this method will be very limited.

---

> > > ### Author Response · Authors · 2022-11-15
> > > **RE: Reviewer ah6w**
> > >
> > > > Extra prior information from the non-LT teacher is not present in the baselines which makes the comparisons unfair (The unbiased teacher will provide useful prior information to the student especially on tail classes).
> > >
> > > $\to$ Thank you for sharing your concern. We cordially refer the reviewer to Tables 1 and 2 where we have presented results for DiVE* where the teacher network used in the original DiVE’s distillation scheme is replaced with **the same teacher** as ours for the suggested comparison. The results clearly show that even with the same teacher as ours, the proposed CANDLE outperforms DiVE* by large margins ($+20.27\%$ for small-scale and $+11.83\%$ for large-scale datasets) on all 15 datasets. We do not plug in the distillation scheme using our teacher for other compared LT methods as they are not distillation based methods; adding a distillation scheme with a teacher network to these methods would result in completely different methodologies altogether.
> > >
> > > In addition, we respectfully argue that utilizing a non-LT teacher for LT recognition itself has not been studied yet and is one of our contributions in this work. Please note that we take one step further in utilizing a non-LT teacher; we calibrate the teacher network in our calibrate & distill framework which leads to $+10.32\%$ gain in average accuracy (see comparison against the Baseline in Tab. 3). This gain suggests that not only is the usage of a non-LT teacher for LT new, but its efficacy is also largely improved by the calibrate & distill framework.
> > >
> > >
> > > > Having a non-LT dataset to train a teacher and then use the proposed method is very challenging.
> > >
> > > $\to$ We appreciate the reviewer sharing the concerns. However, we merely use ImageNet/ImageNet-21K pretrained floating point networks (*e.g.*, EfficientNets) as the non-LT teachers, which are widely available in multiple open source projects. We then calibrate only the classifier of the teacher on the **target LT** data, which is not very challenging (*i.e.*, computationally inexpensive). We refer the reviewer to our first reply to your questions of `How is the teacher trained on non-LT data?` and `While CANDLE can improve efficiency during the inference stage, training seems quite expensive as it needs to train the teacher network first then train a student using CANDLE. Thus, it may decrease the practical benefits of CANDLE.` and Sec. 3.2 for more details on how the non-LT teachers are prepared.
> > >
> > >
> > >
> > > \
> > > **Update on our answer to your question of**
> > > > Assumption of using linear-based classifiers with no discussion on cosine-based classifiers.
> > >
> > > $\to$ We have added results of using the cosine classifiers with CANDLE instead of linear classifiers in A.12 of the second revision and found that linear classifiers show better performance. Please see the general response along with the second revision regarding the added results using the cosine classifiers for more details.

---

> > > > ### Comment · Reviewer_ah6w · 2022-11-19
> > > > **RE: Reviewer ah6w**
> > > >
> > > > "*We respectfully argue that utilizing a non-LT teacher for LT recognition itself has not been studied yet and is one of our contributions in this work.*"
> > > >
> > > > I have a fundamental problem with this approach. It is unpractical. That may explain why the community has not explored this solution. In a realistic scenario and as an example, if I want to train a classifier using CANDLE and the Google Open Images dataset, which is a long-tailed one, it would be very hard to get a non-LT teacher model. Therefore, I cannot use CANDLE in large scale and real long-tailed dataset. Therefore, I question the approach as a viable solution to the long-tailed problem.

---

> > > > > ### Author Response · Authors · 2022-11-26
> > > > > **Re: RE: Reviewer ah6w**
> > > > >
> > > > > We thank the reviewer for the in-depth discussion. We cordially argue that the non-LT teacher is not necessarily pretrained on a *larger* dataset than the target LT dataset. The only reason we used ImageNet1K or ImageNet 21K pretrained teacher (for the small- and the large-scale experiments, respectively) was that the pretrained weights were readily available. As a proof-of-concept for the situation that the teacher is not pretrained on a larger dataset, we could conduct experiments where we pretrain a teacher on a subset of ImageNet1K (*e.g.*, 10~30%) and use it as a teacher for iNat-LT such that the non-LT data is similar or smaller in size than the target LT data.
> > > > >
> > > > > We cordially note that, for the exemplar practical scenario of image classification on the Google Open Image dataset (there are $\sim$9M images with $\sim$21K classes or $\sim$9.7K trainable classes), the same ImageNet21K ($\sim$14M images with $\sim$21.8K classes) pretrained teacher we have used could also be used for this LT classification problem as well. We additionally note that most LT methods  (He at al., 2021; Alshammari et al., 2022; Cui et al., 2021; 2022 and many more) consider Places-LT, ImageNet-LT, and iNat-LT as established large scale LT benchmarks, which we have also followed.

---

### Official Review · Reviewer_xZC3 · 2022-10-25

**Confidence:** 5
**Clarity, Quality, Novelty And Reproducibility:** The authors are expected to give more…
**Correctness:** 3
**Technical Novelty And Significance:** 3
**Empirical Novelty And Significance:** 2
**Recommendation:** 5

**Strength And Weaknesses:**

Strength:
The authors proposed a bianry network for long tailed data, which is interesting. The presentation is good, and easy to follow.

Weaknesses:
1. The authors claimed high variance in binary network can influence the performance of the model, and attempted to lower the high variance. Is there theoretical analysis on this? This seems to be claimed only based on some empirical observation.

2. The proposed method utilized the traditional knowledge distillation for learning a binary network, leading to the limited novelty.

3. In Figure 2, the authors claimed "The classifier weight norms increase at the tail classes, implying high variance". why?

4. Why learn the balance bwtween two losses can be realized by an adversarial learning manner? It is expected to give more explanations.

**Summary Of The Paper:**

The authors proposed a binary network for long tailed data. They transfer knowledge from a calibrated teacher network to the binary network, and utilize adversarial learning for learning the hyperparameters. Experimental results demonstrate the effectiveness of their method.

**Summary Of The Review:**

I incline to reject this paper based on the current issues. However, I will be happy to change my score if the authors can solve my questions.

---

> ### Author Response · Authors · 2022-11-09
> **Answers to the questions of Reviewer xZC3**
>
> > Is there theoretical analysis for the claim of high variance of binary networks and its impact on the performance?
>
> $\to$ Zhu et al., 2019 presented both theoretical (in the supplementary) and empirical studies (Fig. 2 of the main paper) regarding the variance of binary networks. They concluded that binary networks are susceptible to high variance compared to floating point networks. The theoretical study suggests that the usage of the sign function used in the binarization process can incur the binary network to have bigger variance than floating point networks. Their empirical study (Fig. 2) additionally shows this in practice by measuring the output variance of models using various bits from 32 bits (floating point) to 1 bit (binary) when the inputs were perturbed. The results clearly show that the output variance is much higher in binary networks than floating point networks. To reduce the variance of binary networks, they used ensemble techniques and improved the performance. We have added the reference in the introduction of the revision.
>
> - Shilin Zhu, Xin Dong, Hao Su. Binary Ensemble Neural Network: More Bits per Network or More Networks per Bit?, In CVPR, 2019.
>
>
> > Utilizing knowledge distillation for training binary networks is limited in novelty.
>
> $\to$ While we acknowledge that a number of prior arts have used distillation to train binary networks (‘Binary networks’ paragraph in Sec. 2), our usage of distillation differs in the following ways. First, we use an adversarially learned balancing between the loss functions used in distillation (acknowledged as novel and interesting by **F3k8, ah6w**). Second, while previous works have used teacher networks that are trained on similar or same data distributions as the target dataset, we take the off-the-shelf weights pretrained on non-LT data as distillation teachers for a *wide* variety of target LT datasets (acknowledged as interesting and novel scenario by **F3k8, xZC3, ah6w, 2EyL**). We empirically compare our method to the vanilla distillation method (baseline) in Tab. 3; our method has a performance gain of $+16.80\%$ mean average accuracy.
>
>
>
> > In Fig. 2 “The classifier weight norms increase at the tail classes, implying high variance”. Why?
>
> $\to$ We believe the reason why the classifier weight norms increase specifically at the tail classes is due to the data scarcity in the tail classes in LT scenarios (see intro and Sec. 3.1 in the revision); the model fits too much to the few training data.
>
> In addition, the magnitude of the classifier weight norms is closely related to the variance; if the classifier weight norm is large, it incurs high variance (Tikhonov, 1998; Bishop et al., 2006). To elaborate, when the weights’ magnitude is large, even a small difference in the input will result in a large change of the logit value for a particular class (the classifier weight norm that is being multiplied to acquire the logits is large), which results in high variance.
>
>
>
> - Christopher M. Bishop, Nasser M. Nasrabadi. Pattern recognition and machine learning. In Vol. 4, no. 4. New York: Springer, 2006.
>
> - A. N. Tikhonov. Nonlinear Ill-Posed Problems, Springer 1998.
>
>
> > Why learning the balance between two losses can be realized by adversarial learning?
>
> $\to$ Because a simple joint minimization of the balancing weight with the losses themselves would lead to a trivial solution of zeroing out the higher loss value out of the two loss terms $L_{FS}$ and $L_{KL}$, yielding artificially small gradients in the backpropagation (see Sec. 3.3). To avoid this, we first optimize the learned balancing to maximize the combined loss, and then optimize the binary network to minimize the combined loss function. By this, we always optimize the maximum loss value at every iteration (also pointed out by **ah6w**).

---

> > ### Comment · Reviewer_xZC3 · 2022-11-26
> > **RE: Answers to the questions of Reviewer xZC3**
> >
> > Thanks for your replies. After reading the rebuttal, it addressed most of my concerns. However, I still have the concern about the novelty of this paper. Although the authors claimed they used an adversarial way to optimize the loss, and used the off-the-shelf weights pretrained on non-LT data as the teacher model. I think such points can not provide enough contributions to publish in this venue. In addition, after reading other reviewers' comments, I agree Reviewer ah6w that the proposed method might be unpractical, because of the difficulty of obtaining a non-LT teacher model. Thus, the second contribution the authors claimed may be invalid.

---

> > > ### Author Response · Authors · 2022-11-26
> > > **Re: RE: Answers to the questions of Reviewer xZC3**
> > >
> > > Thank you for the further discussion. For the adversarial loss, we respectfully argue that this is a simple but quite effective scheme for better generalization to the *wide* variety of LT datasets, which is important for realistic use cases. For the non-LT teacher, as we mentioned in our response to reviewer **ah6w**, we believe the non-LT teacher is practical as it does not need to be trained on a larger non-LT dataset. Plenty of pretrained weights on sufficiently big non-LT datasets (*e.g.*, ImagetNet21K) exists that should cover most practical LT scenarios. We would greatly appreciate it if the reviewer could follow-up on the on-going discussion with **ah6w**.

---

> ### Author Response · Authors · 2022-11-18
> **Discussion reminder**
>
> We sincerely thank you for your effort in reviewing our submission. We gently remind the reviewer that we tried our best to address your concerns via our replies and revision of the manuscript. As the discussion period is nearing the end, we would be delighted to hear more from you if there are any further concerns.

---

### Official Review · Reviewer_F3k8 · 2022-10-25

**Confidence:** 2
**Correctness:** 3
**Technical Novelty And Significance:** 2
**Empirical Novelty And Significance:** 2
**Recommendation:** 3

**Clarity, Quality, Novelty And Reproducibility:**

The clarity and reproducibility of the paper are very low and it's the main weakness.
I am missing proper definitions of major relevant terms, such as
* long-tailed data distribution
* classifier weight norms
* imbalance ratio
* variance at classes
* multi-resolution input
* encoders
* $Atten_\phi$

Further, the issue of optimization is largely glossed over. The training of binary networks can be done in multiple ways and is non-trivial. Here, it's unclear how the binary weights are trained. Further, the authors propose a min-max optimization, which typically also requires some finesse. I read somewhere that Adam is used but I doubt that you can just through an optimizer on this objective and get decent results. How is the step size set? Some choices for the objective are unmotivated, for example, the choice for the cosine distance loss for the encoders. How are hyperparameters tuned? I didn't understand at all how the multi-resolution input is integrated and how this is handled in the training.

I am not familiar enough with the field that I could assess the novelty. Quality is also quite unclear due to the clarity issue.

**Strength And Weaknesses:**

# Strengths
* I like that the weight of the linear combination of losses is optimized
* Figures and plots are used to illustrate various points
* Dataset statistics are given in the appendix
* The considered task of long-tail class distributions under resource constraints is interesting and possibly impactful for many applications
# Weaknesses
* The paper is not clear. Many terms are never properly defined, there is a lack of mathematical definitions, big chunks of the method are a black-box to me, I don't understand most Figures and the reasoning is also often unmotivated.
* The writing is often repetitive in what the authors are going to do without actually explaining it, which makes for a frustrating read
* The related work just lists related papers without actually pointing out how the existing work builds the fundament for benchmarks.

**Summary Of The Paper:**

The authors consider the task of classifying data with long-tailed distributions of class samples under resource constraints. The resource constraints are addressed by using binary neural networks. The authors propose to use floating point networks pretrained on standard benchmarks and to retrain the classifier layer on the long-tailed data (here called _calibration_). The proposed objective is a linear combination of the loss of the binary network and the retrained floating point network, and the floating point and binary encoders. The weight of this linear combination is learned in an adversarial way, such that the part of the objective with a higher loss gets a higher weight. In addition, a mult-resolution module is added. The experiments indicate high accuracy of the proposed method in comparison to competitors on various datasets for which a long-tail class distribution has been created.

**Summary Of The Review:**

Possibly a good paper that is not acceptable at this point in my view, because of its unclarity.

# After Rebuttal Thoughts
I see that the authors revised the paper carefully and that they provided more clarity regarding the terminology and methodology. However, in order to assess the revised version, I would have to re-read the whole paper. Unfortunately, I don't have time to do so, hence, I'll decerease my confidence score and give other reviewers the floor.

---

> ### Author Response · Authors · 2022-11-09
> **Answers to the questions of Reviewer F3k8 (2/2)**
>
> > Related work just lists relevant work.
>
> $\to$ Thank you for the suggestion! To better explain how the previous works in binary networks build the fundamentals, we revised the categorization of the prior arts into architectural improvements, architecture search, training techniques, and others in the revision. We present a more in-depth discussion regarding recent works on training binary networks that use pretrained FP networks as it is more relevant to our work.
>
> For the long-tail recognition literatures, we believe that we have explained how the prior arts in LT build the fundamentals for benchmarks by categorizing them into different methods such as class rebalancing, logit adjustment, two stage training, and knowledge distillation by pointing relevant prior arts for each of the categories. We also have discussed prior arts that are more relevant to our work in greater detail, such as prior two-stage training or methods that increase model capacity for LT.
>
> We hope that the revised manuscript gives a better explanation of the prior arts. But if you find it needs more elaboration, please let us know. We’ll apply that in the following revision.
>
>
> > Writing is often repetitive.
>
> $\to$ Thank you for the comment. We try to reduce the repetitions and hope that the revised manuscript is less repetitive. In addition, if the reviewer still finds repetitive texts, we would be happy to fix the writing per the reviewer’s feedback.

---

> ### Author Response · Authors · 2022-11-09
> **Answers to the questions of Reviewer F3k8 (1/2)**
>
> > Missing proper definitions of major relevant terms.
>
> - Long-tailed data distribution
>
>   $\to$ It refers to data distributions where there are a few classes with the majority of samples (called ‘head’ classes) and the majority of classes have few samples (called ‘tail’ classes). We use the same definition following numerous prior arts (Cao et al., 2019; Kang et al., 2019; Cui et al., 2021; 2022 to name a few).
>
> - Classifier weight norms
>
>   $\to$ They refer to the norms of the weights of the final fully-connected layer as is also used in Kang et al., 2019; Alshammari et al., 2022. We add the details in the introduction of the revision to make this clearer.
>
> - Imbalance ratio
>
>   $\to$ It is the ratio of $\frac{\text{number of maximum samples in the head classes}}{\text{number of minimum samples in the tail classes}}$. It is frequently used in the long-tail recognition literature (Cao et al., 2019; Kang et al., 2019; Cui et al., 2021; 2022 to name a few).
>
> - Variance at classes
>
>   $\to$ We mean the variance (in the bias-variance trade-off) of the classifier weight norms for a specific class.
>
> - Multi-resolution input
>
>   $\to$ We resize the input images to be $128\times128, 224\times224, 480\times480$ and use the resized images sequentially as the multi-resolution inputs to the teacher FP network. We clarify this in Sec. 3.4 of the revision.
>
> - Encoders
>
>   $\to$ It refers to the feature extraction part (*i.e.*, layers before the classifier - the multi-layer perceptron (MLP)) of the floating point and binary networks respectively. We add this detail in Sec. 3.2 of the revision.
>
> - $Atten_\phi$
>
>   $\to$ It is a MLP that takes in two scalar values, $L_{KL(x)}$ and $L_{FS(x)}$, and outputs a single scalar value $\lambda_\phi$ (see Eq. (3) and A.5). In particular, we pack the two scalar values into a 2-dimensional vector, and provide it to the MLP. Then, the MLP outputs a single scalar value which is passed through a sigmoid function to obtain a value between $0$ and $1$, which we call $\lambda_\phi$. We add these details to A.5 in the revision.
>
>
> > How are binary networks trained?
>
> $\to$ The forward pass is done by binarized convolutional operations (Rastegari et al., 2016) and the backward pass is done by the straight through estimator (STE) (Courbariaux et al., 2016). Please refer to A.1 for the details.
>
>
> > Min-max optimization details (finesse) such as step size.
>
> $\to$ We perform a single maximization step per every minimization step and found that this is sufficiently effective. This is because unlike other complicated min-max optimization schemes such as GAN, we optimize only a scalar balancing factor $\lambda_\phi$ which does not require much finesse. We have added the details in A.5 of the revision.
>
>
> > Choice of cosine distance loss for the encoders is unmotivated.
>
> $\to$ The motivation to use the cosine distance for the distance between the feature vectors is that it calculates how the feature vectors are aligned w.r.t. the angles only and not the magnitude, following the self-supervised learning literature (Chen et al., 2020; He et al., 2020). We have added this in Sec. 3.2 of the revision.
>
> - Ting, Chen, Simon Kornblith, Mohammad Norouzi, and Geoffrey Hinton. A simple framework for contrastive learning of visual representations. In ICML, 2020.
> - Kaiming He, Haoqi Fan, Yuxin Wu, Saining Xie, and Ross Girshick. Momentum contrast for unsupervised visual representation learning. In CVPR, 2020.
>
>
> > How are the hyper-parameters tuned?
>
> $\to$ The only hyper-parameter we tune is the initial learning rate. We tried a number of values for it, *i.e.*, $(0.001, 0.005, 0.01)$, but found that the performance difference is marginal in most of the 15 datasets in our empirical validations. Thus, we use the fixed value of $0.01$ as the initial learning rate for all experiments. Note that further tuning of it may lead to additional performance improvement. We have added the details in A.5 of the revision.
>
>
> > How are the multi-resolution inputs handled?
>
> $\to$ We group the multi-resolution inputs with the same resolution (*e.g.*, $128\times128$, $224\times224$, and $480\times480$) into separate batches and perform the forward pass with just the teacher network in both the calibration and distillation stages to obtain multi-resolution features. The multi-resolution features from the teacher network are spatially resized using interpolation to be of equal dimensions and then channel-wise concatenated. The feature distillation loss uses the channels of the concatenated feature vector corresponding to the $224\times224$ resolution input and the teacher network’s classification uses all the concatenated channels. Please see Fig. 3 and A.5 for details.

---

> ### Author Response · Authors · 2022-11-18
> **Discussion reminder**
>
> We sincerely thank you for your effort in reviewing our submission. We gently remind the reviewer that we tried our best to address your concerns via our replies and revision of the manuscript. As the discussion period is nearing the end, we would be delighted to hear more from you if there are any further concerns.

---

### Author Response · Authors · 2022-11-09
**General response**

We thank the reviewers for their helpful feedback and encouraging comments such as learning binary networks on long-tailed data being interesting (**F3k8, xZC3**), novel (**2EyL**), important (**ah6w**) and practical (**F3k8, 2EyL, ah6w**), the balancing of the losses being learned is new (**F3k8**) and interesting (**ah6w**), plots and figures being used to illustrate various points (**F3k8**), the presentation being good (**xZC3**), the high-level ideas being explained well (**ah6w**) with interesting analysis (**2EyL**), and the empirical results showing high accuracy (**F3k8, ah6w**) and the effectiveness (**xZC3, 2EyL**) of the proposed method.

We have uploaded the first revised version of the manuscript. Note that the limitations paragraph has been moved to section A.12 in the appendix for sake of space to accommodate the changes made in the revision.

---

### Author Response · Authors · 2022-11-15
**General response with the second revision**

Note that the second revision of our manuscript has been uploaded.

Summary of the changes: We added the results using cosine classifiers (**ah6w**, **2EyL**) in A.12. The results show that using cosine classifiers is slightly worse than using the linear classifier (as is proposed in CANDLE). Specifically, by using a linear classifier, taking the mean of the results on the 12 small-scale datasets, we observe a **$+4.25\%$** gain in average accuracy and **$+10.21\%$** gain in tail class accuracy over a cosine classifier. The results empirically support our intuition that while cosine classifiers may be effective for LT recognition with FP networks, they may not be as effective for binary networks as for FP networks. We believe it is because the weight norms’ magnitudes are not as useful as a discriminative statistic with a cosine classifier in supplementing the limited capacity of binary networks.

---

### Author Response · Authors · 2022-11-18
**Notification of the third revision**

Note that the third revision of our manuscript has been uploaded.

**Summary of the change**: We added a brief discussion regarding Adam and how it affects the weight norm trend in LT that originated from the additional discussions with reviewer **2EyL** along with minor changes in the captions for Figures 1 and 2.

---

### Decision · Program_Chairs · 2023-01-20

**Decision:**

Reject

**Justification For Why Not Higher Score:**

lack of clarity and justification on several points

**Justification For Why Not Lower Score:**

N/A

**Metareview: Summary, Strengths And Weaknesses:**

### Description

The goal of the paper is to train a binary neural network to the best accuracy for a given imbalanced (long-tailed) dataset. The test distribution is assumed to be the same. For the application of the method this is the only dataset considered, so I will refer to it as just the dataset.

The method of the paper is the following pipeline:
1) Start with a pretrained full precision network, throw away the classifier part.
2) Refit the classifier part of the network to the dataset.
3) Train the binary network with a variant of distillation to mimic the full precision teacher in 2) in logits and features at the last layer.

The novelty of the paper, besides the pipeline itself, is around several points:
* experimental diagnostics of long-tail performance issues through the perspective of weight norms
* training procedure (adversarially learned balance)
* efficient multi-resolution distillation

### Decision

Reviewers have positively evaluated that the problem addressed is important and practically relevant, that the method outperforms baselines by a large margin, that the training procedure is interesting. However there were several important points not fully clarified by the rebuttal and further issues (all detailed below). We carefully discussed these issues and recommend that the paper is revised and resubmitted.

### Clarity / Soundness issues

1) The paper argues a lot about the high variance (e.g. Section 3.1 HIGH VARIANCE FOR BINARY NETWORKS IN LONG -TAILED DISTRIBUTION). The notion of variance used in different places seems to be different and is never made clear (see meeting summary).
It would be appropriate to define exactly variance of what and with respect to which source of randomness. The implications such as "large weight magnitude" implies "high variance" should be substantiated in the paper. The paper should elaborate (or review the prior work) on why large weights are indicative of overfitting and poor generalization for rare classes. Additionally, in contrast to the submission, plots in Alshammari et al., (2022) show a reverse dependence: common classes have large weight magnitude than rare ones, adding to the confusion.

2) The importance of having the pretrained teacher on a non-LT data is not clear. Authors argue that it is a practical setup and such networks are readily available. The reviewer ah6w argues that this is not practical since in the large-scale setup we rather have LT datasets collected. In the meeting we converged rather to the interpretation that the assumption of non-LT pretrained model is not important. It only provides useful embeddings and as the classifier part of the network is discarded, it does not retain specific information about the rare classes. In that case the paper should support this by evidence and not emphasize the pretraining on a **balanced** dataset. Any pretrained model, e.g. unsupervised, semi-supervised, self-supervised, should be good. It would be expected to bring gains to the training of binary neural networks on any data, not necessarily LT data.
See more details in the "Summary Of AC-reviewer Meeting". Please contact PCs if you don't get to see these comments.

3) Another major concern is the justification of the adversarially learned balance procedure. The optimally trained $\phi$ should be equivalent to
$\max_{\lambda \in [0,1]} (\lambda L_{KL} + (1-\lambda) L_{FS}) = \max(L_{KL}, L_{FS})$
So from the viewpoint of the design of the objective, this is just equivalent to considering $\max(L_{KL}, L_{FS})$ as the learning objective. The latter is not sound, because $L_{KL}$ and $L_{FS}$ are different kinds of losses on different number of units, and need a weight coefficient to make the comparison meaningful. The procedure is therefore does not appear rational. Its effect is some indirect schedule of changing $\lambda$ due to learning dynamics. The fact that eventually $\lambda$ converges to $1$ means that towards the end only the encoder is improved but not the classifier.  This schedule is controlled not by a hyperparameter but by e.g. the design of the Attention network and the optimizer. This is very obscure from the point of view of a rational design. If, nevertheless, it is a good heuristic in practice for some reason, more experimental evidence is needed to evaluate, explain and justify it.

**Summary Of Ac-Reviewer Meeting:**

Points discussed:

* Which variance is meant in the paper and why it is identified/ associated with high weight norm. We concluded it is confusing in the paper and is sometimes identified with the norm of the weights of a classifier associated to a given class and sometimes distinguished (e.g. "the classifier weight norms increase at the tail classes, implying high variance"), without clearly specifying variance of what and with respect to which source of randomness. The high variance issue in binary networks (Zhu et al., 2019) cited is the variance of predictive distribution w.r.t. perturbation of the input, and is not relevant for the paper. In the context "bias and variance of a model", it is apparently a different notion: good (sharp) predictors are expected to have high variability of predictions. It is not well-motivated why high weight norms are bad per se and needs to be reduced.

* Why there is a reverse trend in the weight norm head-to-tail in comparison to Alshammari et al., 2022? It appears to be due to Adam optimizer versus SGD used as confidemend by the authors if Fig. 8 of appendix. However, since the trend is reverse, it is not clear whether the motivation to reduce the weight norm is valid in general.

* Non-LT pretrained model interpretation. We considered two possibilities:
  1) The assumption of non-LT pretrained model is important. In particular if it gets to see more examples of the rarer classes than in the target LT dataset, then it is a serion flow. In the extreme case the following experiment would show a perceived "improvement": take a balanced data set for pretraining, then subsample it to obtain LT dataset and measure the performance on that subsampled dataset. This clearly benefits from pre-training but does not have a practical meaning.
  2) The assumption of non-LT pretrained model is not important. It only provides useful embeddings and as the classifier part of the network is discarded, it does not retain specific information about the rare classes. In that case the paper should not emphasize the **non-LT** pretrained model concept. Any pretrained model, on a generic, weakly related dataset or using unsupervised methods e.g. (some contrastive representation learning) should be good. In that case the paper should be revised accordingly, possibly with additional experiments with such pretrained models.

We agreed, including ah6w, that it is rather the second option, but the situation is unclear for the readers. Reviewers recommended that using CLIP features or self-supervised pretrained models would be beneficial (and would help clarify this question), e.g. Kang et al. (2021) Exploring Balanced Feature Spaces for Representation Learning.

* Reviewers with the expertise in the LT problem, noticed that the total accuracy is an inadequate metric and that fitting a classifier for a FP network itself does not take into account the LT issue and can result in a bias later on transferred to the binary network. This was well addressed by the rebuttal revision adding Table 8 and Figures 9-10. It should be better discussed in the main text and possibly the accuracy breakdown can be shown in the main text.

* We discussed the adversarially learned balancing method. There was a consensus on that this technique is at least not clear if not misleading.

We then discussed advantages of the work. The work is towards applying binary networks in realistic scenarios with LT dataset, which is very practical and important. It is first to study such scenario and exposes interesting observations and challenges and proposes a solution which shows a substantial improvement. At the same time lack of clarity implies low(er) reproducibility. After the meeting the overall assessment was more positive but we came to the conclusion that a substantial revision is still rather necessary.